



# Permafrost thaw couples slopes with downstream systems and effects propagate through Arctic drainage networks.

Steven V. Kokelj[1], Justin Kokoszka[1,2], Jurjen van der Sluijs[3], Ashley C.A. Rudy[1], John Tunnicliffe[4], Sarah Shakil[5], Suzanne Tank[5], Scott Zolkos[5,6]

[1]Northwest Territories Geological Survey, Yellowknife, NT, X1A 2L9, Canada

[2]Wilfrid Laurier University, Yellowknife, NT, X1A 2L9, Canada

[3]Northwest Territories Centre for Geomatics, Yellowknife, NT, X1A 2L9, Canada

[4]Department of Earth Sciences, University of Auckland, Auckland, NZ

[5]Department of Biological Sciences, University of Alberta, Edmonton, AB, T6G 2E3, Canada

[6]Woods Hole Research Centre, Falmouth, MA, 02540, USA

*Correspondence to*: Steven V. Kokelj (steve_kokelj@gov.nt.ca)



## Abstract.

The intensification of thaw-driven mass wasting is transforming glacially-conditioned permafrost terrain, coupling slopes with aquatic systems and triggering a cascade of downstream effects. Within the context of recent, rapidly evolving climate controls on the geomorphology of permafrost terrain we: A) quantify three-dimensional slump enlargement and describe the processes and thresholds coupling slopes to downstream systems; B) investigate catchment-scale patterns of slope thermokarst (thaw slumps and slides) impacts and the geomorphic implications; and C) project the propagation of effects

through hydrological networks draining continuous permafrost of northwestern Canada. Power-law relationships between thaw-slump area and volume ($R^2 = 0.90$), and thickness of permafrost thawed ($R^2 = 0.63$), combined with the multi-decadal (1985-2018) increase in areal extent of thaw-slump disturbance show a two-order of magnitude increase in catchment-scale geomorphic activity and the coupling of slope and hydrological systems. Predominant catchment effects are to first- and second-order streams where sediment delivery commonly exceeds stream transport capacity by orders of magnitude

indicating millennial-scale perturbation of downstream systems. Assessment of hydrological networks indicates thaw-driven mass wasting directly affects over 6,760 km of stream segments, 890 km of coastline and 1,370 lakes in the 994,860 km$^2$ study area. Downstream propagation of slope thermokarst indicates a potential increase in the number of affected lakes by at least a factor of 4 (n >5,600), impacted stream length by a factor of 7 (>48,000 km) and defines several major impact zones to lakes, deltas and coastal areas. Prince of Wales Strait is the receiving marine environment for greatly increased sediment

and geochemical fluxes from numerous slump impacted hydrological networks draining the landmasses of Banks and Victoria Islands. Peel and Mackenzie Rivers are globally significant conveyors of the slope thermokarst cascade delivering effects to North America's largest delta and the Beaufort Sea. Climate-driven erosion of ice-rich slopes in permafrost-preserved glaciated terrain has triggered a time-transient cascade of downstream effects that signal the renewal of post-glacial landscape evolution. Glacial legacy and the patterns of continental drainage dictate that terrestrial, freshwater, coastal,

and marine environments of western Arctic Canada will be an interconnected hotspot of thaw-driven change through the coming millennia.





## 1 Introduction

Climate-induced permafrost thaw will drive the geomorphic evolution of circumpolar landscapes (Kokelj and Jorgenson, 2013), and terrestrial, freshwater and coastal ecosystems (Vonk et al., 2019). Thawing of ice-rich, glacially-conditioned permafrost terrain (Kokelj et al., 2017a) is rapidly mobilizing vast stores of previously frozen materials, reconfiguring slopes and impacting downstream environments (Fig. 1) (Balser et al., 2014; Rudy et al., 2017a; Tank et al., 2020). These and other similar results highlight the need to quantify slope thermokarst intensification in a robust physical framework, better understand the rapidly evolving linkages between thawing slopes and downstream environments, and predict the propagation of effects across watershed scales.

The nature of permafrost thaw and downstream consequences will increasingly define trajectories of Arctic change. Study of thawing slopes and shorelines (Lacelle et al., 2010; Ramage et al., 2017; West and Plug, 2008), and characterization of permafrost physical and geochemical properties (Lacelle et al., 2019), informs the projection of downstream cumulative-effects and implications for carbon and contaminant mobilization (Littlefair et al., 2017; Ramage et al., 2018; St. Pierre et al., 2018; Tank et al., 2020). Recent, rapid increases in the areal extent of upland thermokarst (Lewkowicz and Way, 2019; Segal et al., 2016a; Ward-Jones et al., 2019), and shifts in hydrological, sedimentary and geochemical regimes (Abbott et al., 2015; Kokelj et al., 2013; Littlefair et al., 2017; Malone et al., 2013) can be linked with significant aquatic ecosystem impacts (Chin et al., 2016; Houben et al., 2016; Levenstein et al., 2018; Thienpont et al., 2013). The processes and feedbacks driving the recent evolution of thawing slopes (Kokelj et al., 2015; Zwieback et al., 2018) has received less attention. Thaw-driven landslide effects on permafrost catchments have been investigated through nested watershed studies (Beel et al., 2018; Shakil et al., 2020a; Zolkos et al., in review), and the cascade of effects have been inferred through geochemical trend analyses of large Arctic rivers (Tank et al., 2016; Zolkos et al., 2018). The broad-scale distribution of slope thermokarst determined through empirical (Kokelj et al., 2017a), remote sensing (Brooker et al., 2014; Nitze et al., 2018; Olefeldt et al., 2016) and modelling approaches (Rudy et al., 2017b) have typically not been designed to elucidate physical processes, downstream connectivity and attendant effects. Despite the growing geomorphic and geochemical influence of thaw-driven mass wasting on terrestrial, aquatic and marine systems (Vonk et al., 2019), and the potential for rapid carbon release (Turetsky et al., 2020), fundamental knowledge gaps persist in our understanding of climate-driven amplification of slope thermokarst, the evolution of downstream linkages and cascade of consequences.

Anticipating the cumulative effects of slope thermokarst intensification on permafrost landscapes and downstream environments requires a better knowledge of thaw-driven geomorphic processes and the evolution of connectivity across a range of spatial scales. First, exploring the geomorphic and geochemical implications of slope disturbance requires determining the relationships between disturbance area, volume, and depth of permafrost thawed so that changes in landslide count or areal extent (Lantz and Kokelj, 2008; Lewkowicz and Way, 2019; Ward-Jones et al., 2019) can be considered in a more robust physical framework (van der Sluijs et al., 2018). To advance understanding of processes and feedbacks driving



the rapid evolution of slope thermokarst disturbance and hillslope channel coupling, including the patterns and rates of thaw-
driven disturbance enlargement, downslope transport of material derived from erosional features, and quantification of slope
sediment budgets, conventional satellite- and airborne-derived planform information on slope mass-wasting should be
quantified with high-resolution 3D survey techniques, such as Light Detection and Ranging (LiDAR), or drone-based
Structure-from-Motion (SfM). Secondly, analysing thermokarst effects within the context of a drainage network is required
to advance the understanding of slope-to-stream linkages and downstream connectivity (Wohl et al., 2019). Describing slope
thermokarst within a hydrological framework will enable the description of watershed effects, and provide a platform for
modelling the transient propagation of geomorphic and geochemical impacts. Finally, considering thermokarst distribution
and trajectories of change within an appropriate theoretical or geomorphic context (Ballantyne, 2002) will support the
explanation of spatial patterns, rates and magnitudes of geomorphic change (Kokelj et al., 2017a), and geochemical effects
(Lacelle et al., 2019; Tank et al., 2020), and the contextualization of thermokarst field investigations, as well as conceptual
and physically-based models (Turetsky et al., 2020).

To address these knowledge gaps, and specifically, to better understand the (A) processes that drive the intensification of
thaw-driven mass wasting and slope to stream coupling, (B) the distribution of catchment effects, and (C) their propagation
across watershed scales, we present a suite of spatially nested case-studies bounded by Arctic drainage from continuous
permafrost of northwestern Canada (Fig. 1). This $10^6$ km$^2$ study region contains a wide range of climate and permafrost
temperature regimes (Smith et al., 2010), ground ice conditions (O'Neill et al., 2019), biophysical gradients (Lantz et al.,
2010), and geological environments, including unglaciated and glaciated terrain (Dyke and Prest, 1987; Dyke et al., 2003).
The latter is of central relevance to study design, in particular where permafrost has maintained ice-marginal moraine,
glaciolacustrine, glaciofluvial and glaciomarine deposits in a quasi-stable state, preserving relict ground ice (Lakeman and
England, 2012; Mackay, 1971; Murton et al., 2005; Pollard, 2000) and constraining slope evolution through a cooling
Holocene (Porter et al., 2019). Today, these environments host numerous areas of intense thaw-driven geomorphic change
(Fig. 1) where landslides, predominantly in the form of retrogressive thaw slumps, or retrogressive thaw-flow slides, are
mobilizing slope materials, transforming landscapes (Kokelj et al., 2015), and triggering an array of downstream effects
(Chipman et al., 2016; Rudy et al., 2017a; Zolkos et al., 2018). The focus of fine-scale investigations in this study mainly
considered retrogressive thaw slumping because it is a primary mode of thaw-driven slope failure in the study region (Segal
et al., 2016a), and often a modifier of slopes subject to other types of landslides. Our broad-scale mapping included multiple
slope failure modes, including thaw slumps, and shallow and deep translational slides, so we utilized more inclusive
terminology such as slope thermokarst, or thaw-driven landslides and mass wasting when discussing these results.

To quantify the enlargement rates of retrogressive thaw slumps, assess allometric relationships (i.e. area and volume) (cf.
Bull, 1964), and to explore thresholds that govern slope-to-stream connectivity (Objective A) we analysed fine-scale
topographic data derived from LiDAR for a large population of thaw slumps from across biophysical transitions on the Peel



Plateau (Kokelj et al., 2017b), and Anderson Plain and Tuktoyaktuk Coastland areas (Rampton, 1988), and from repeat Unmanned Aerial Vehicle (UAV) terrain surveys of individual disturbances on the Peel Plateau (Fig. 1). To investigate catchment-scale patterns of slope thermokarst effects (Objective B), we combined thaw-slump mapping with empirical models to estimate disturbance volume, assess fluvial patterns of thermokarst effects and derive first-order estimates of slope

denudation for a medium-sized catchment (Willow River, $10^3$ km$^2$) in the Mackenzie Delta region (Fig. 1). Detailed investigations of slope thermokarst disturbance distribution within a fluvial network were extended to a range of catchment scales for the northern Peel Plateau (3,520 km$^2$) and compared with conditions from southeastern Banks Island (1,220 km$^2$), and patterns of fluvial-geomorphic effects were explored by contrasting stream sediment fluxes across catchment scale and disturbance status. To assess the potential propagation of slope thermokarst across broad spatial scales (Objective C),

watershed-scale assessments ($10^4$ to $10^5$ km$^2$) of slope thermokarst effects were analysed within a Strahler-order framework for the Banks Island, Amundson Gulf, Peel River and Keele/Redstone watersheds. Finally, slope thermokarst effects on stream, lake and coastal environments were integrated through a flow accumulation analysis to project the potential cascade of thermokarst effects through hydrological networks of northwestern Canada. With these data and analyses, we link the intensification of thaw-driven slope processes with the coupling of downstream systems and the propagation of effects

through hydrological networks to show the interconnected nature of slope thermokarst hotspots, and to project the patterns of emerging sedimentary and geochemical regimes that will define major Arctic change through the coming millennia.



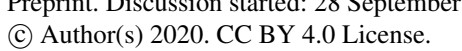


**Figure.1.** Study region map showing the distribution and dominant geomorphic environments affected by thaw-driven mass wasting, and the locations and scales of investigation constrained by the 994,860 km$^2$ area of Arctic drainage from continuous permafrost of northwestern Canada. Fine-scale thaw slump mapping utilizing high resolution UAV and LiDAR terrain models is indicated by the orange corridors (Peel Plateau and Tuktoyaktuk Coastlands and Anderson Plain); small to medium scale catchments including Willow River, Peel Plateau and southeastern Banks Island areas are indicated by red polygons; major watersheds are outlined in blue and Arctic drainage from continuous permafrost which defines the broadest scale of investigation is shaded in grey. The disturbance data on the map is adapted from Segal et al., 2016b and Kokelj et al., 2017a. Late glacial limit is from Dyke and Prest, 1987, bedrock geology is from Fulton, 1995 and the permafrost boundary is from Brown et al., 1997. Base map is from ESRI ArcGIS Online.




## 2 Study area and methods

### 2.1 Study Area

Arctic drainage from continuous permafrost of northwestern Canada comprises an area of about 1 million km$^2$, characterized by diverse permafrost, geological, climate and ecosystem conditions (Fig. 1). The Mackenzie and Peel River systems drain the south central and western parts of the study area. Several small hydrological systems characterize the Yukon Coastal plain, whereas middle to large northward flowing rivers drain tundra of the interior platform and shield terrain northeast of the Mackenzie Basin. An abundance of small to medium sized streams comprises tundra watersheds of Banks and Victoria

Islands. This large study area includes warm permafrost in the southern and western regions where mean annual ground temperatures in undisturbed forested and alpine tundra terrain range from above -1°C to about -3°C (Smith et al., 2010; O'Neill et al., 2015), a transition to low Arctic tundra where permafrost is typically greater than 100 m in thickness with mean annual ground temperatures below -4°C (Kokelj et al., 2017c), and Arctic tundra of Banks and Victoria Islands where frozen ground is hundreds of metres thick and typically below -10°C (Smith et al., 2010). Abundant ice-rich terrain in the

region is indicated by the widespread occurrence of segregated ice, relict ground ice, and ice-wedge ice, (O'Neill et al., 2019 and references within). The distribution and abundance of these main ground-ice types are associated with the legacy of glaciation, distribution of fine-grained, frost-susceptible sediments and a cold Holocene climate. The spatial distribution of retrogressive thaw slumps in this region confirms the abundance of ice-rich terrain and thaw sensitive slopes, and the broad-scale association with ice-marginal, permafrost preserved glacigenic terrain (Fig. 1) (Kokelj et al., 2017a). The unglaciated

western margins of the study region are mountainous and permafrost is generally ice-poor. Numerous high energy streams and rivers draining the Cordillera have incised ice-rich unconsolidated glacial deposits along their course to the Mackenzie or Peel Rivers. These thick deposits of fine-grained tills and glacigenic materials, derived primarily from sedimentary bedrock, define the western margins of the Laurentide glacial limits and the contrast with the eastern part of the study region, which is characterized by extensive areas of shield bedrock veneered by patches of glacial materials derived from

Precambrian rock (Dyke and Prest, 1987). The predominant form of slope thermokarst across the region is retrogressive thaw slumps (Kokelj et al., 2017a), with shallow landslides and deep seated translational failures having local importance that increases southward, particularly in areas of greater relief (Aylsworth et al., 2000). To examine processes driving intensification of slope thermokarst and the patterns of effects across hydrological networks we applied multiple methods involving field study and mapping at slope, catchment, and watershed scales described in the following sections. Datasets

supporting this work include (Kokoszka and Kokelj, 2020; Rudy and Kokelj, 2020; Rudy et al., 2020; Shakil et al., 2020b).

### 2.2 Fine-scale topographic data to explore hillslope-channel coupling

To describe slope-scale processes of thaw slump development and slope-to-stream connectivity, terrain surveys by airborne LiDAR from fixed-wing aircraft and SfM photogrammetry from UAV were used to derive a sequence of Digital Terrain



Models (DTMs) (van der Sluijs et al., 2018) covering an 8-year span from 2011 to 2018. These surveys comprised a series of repeat DTMs enabling total and annual volumes of material displaced by thaw-driven mass wasting to be estimated by: 1) calculating differenced DTMs (DOD); 2) determining uncertainties and masking DODs based on minimum levels of change detection thresholds; and 3) summarizing the cell values for the scar-zone (erosion) and debris tongue (deposition). Full data acquisition and processing methods are provided in van der Sluijs et al (2018). Survey metadata is provided in Table S1.

Overlapping UAV photos were acquired from a continuum of slump features of varying sizes on the Peel Plateau (Fig. 1) between 2015 and 2018, along with ground control data acquired through differential Global Navigation Satellite System (GNSS) surveys.  Aerial photo datasets were processed into georeferenced colour orthomosaics and point clouds using SfM software packages (Smith et al., 2016). Noise-filtered and ground-classified point clouds were rasterised in 0.5-m spatial resolution bare-earth DTMs and resampled bilinearly to 1 m spatial resolution for spatial consistency with LiDAR models.

The LiDAR data with an average point density of 1.7 $m^{-2}$ was collected on 25-28 August 2011 by vendor McElhanney Consulting Services Ltd. (Vancouver, BC, Canada) along two study corridors (Fig. 1). The first is a 162 km long, 6-9 km wide portion of the Dempster Highway across the Peel Plateau comprising an area of 1,032 $km^2$ (O'Neill et al., 2015; Kokelj et al., 2017b). The second corridor is a 9-19 km wide and 139 km long area 1,478 $km^2$, comprising the Inuvik-to-Tuktoyaktuk Highway corridor which crosses the Anderson Plain and Tuktoyaktuk Coastlands (Rampton, 1988). Following

initial processing by the vendor, a baseline 1-m LiDAR DTM was created in ESRI ArcGIS 10.4.1 ("LAS Dataset to Raster" tool) using mean ground point elevations and Delaunay triangulation with linear interpolation to fill data voids. Vertical datum differences between the LiDAR and UAV surveys were corrected to ensure data compatibility.

To explore the associations between thaw slump enlargement and slope-to-stream coupling, we quantified the volumes of slump erosional concavities and debris tongue deposits for a slump-size continuum exemplified in five disturbances in the

Peel Plateau region where DTM data was derived from 2018 UAV surveys. Notably, geomorphic activity at one of the thaw slumps (CB) had accelerated significantly throughout the monitoring record (2011-2018) (Table S1). The 2011 LiDAR DTM provided a topographic baseline that could be used to reconstruct pre-disturbance terrain surfaces so that volumetric changes associated with thaw-driven slope erosion and downstream deposition could be estimated. The pre-disturbance surfaces were manually reconstructed using LiDAR-derived 2-m contours aided by historical aerial photographs, and circa 1970 Canadian

Digital Elevation Model (CDEM) (Government of Canada, 2000), following van der Sluijs et al. (2018). Pre-disturbance valley morphology required valley-bottom elevations to be constrained, so undisturbed stream elevations were sampled at 10 m intervals up to 200 m above and below debris tongue deposits. Fitted polynomial curves were used to model pre-disturbance valley bottoms, constraining contour line reconstructions, which were interpolated to a 1-m DTM using the ArcGIS 10.4.1 "Topo-to-Raster" tool. Total scar zone and debris tongue volumes were estimated, and DODs provided a

volumetric estimate of the year to year changes.





## 2.3 Regional Scale Data Acquisition and Processing

To explore relationships between planform thaw slump scar area and volumetric erosion we digitized scar areas, and estimated the volume and depth of the erosional concavities for 71 thaw slumps from the 2011 LiDAR corridors described
above (Fig. 1). Utilizing LiDAR hill shaded DTMs, all active or recently-active scar and debris tongue areas were digitized at a 1:2,000 scale. Each polygon defined either a scar zone where the debris tongue was indistinguishable or a scar-only area where a debris tongue could be identified as a geomorphic feature distinct from the active scar zone. Slumps were designated as being associated with fluvial, lacustrine or coastal systems, with an assigned integer to indicate the downstream connectivity, where "0" is no connection with the downstream environment; "1" is connection between the bare scar area
and downstream environment, and "2" indicates evidence of downstream deposition, including a debris tongue in a valley bottom, or a sediment lobe protruding into adjacent lake or coastline.

For developing regional-scale slump DTMs and DODs we followed general procedures described in Sect. 2.2, with some adaptations to streamline processing of a large sample population. Firstly, only "classic" cuspate or bowl shaped thaw slump
forms were targeted for analysis: more complex elongated and polycyclic features, common along shorelines, introduced greater uncertainty in the process of automated pre-disturbance terrain reconstruction. Several parameter adjustments in ArcGIS LAS Dataset to Raster tool were implemented to minimize vegetation influences. Rather than linearly triangulating mean elevation of ground-classified points, minimums were binned into a 1-m grid to better represent ground elevation in (re-)vegetated terrain (Gould et al., 2013; Meng et al., 2010). Reconstructing pre-disturbance slump topography was
automated by removing all points in the slump scar and interpolating the pre-disturbance surface with points adjacent to the scar. Natural neighbour void-filling interpolation was applied as a balance between accuracy and shape reliability across a range of natural environments (Bater and Coops 2009; Boreggio et al., 2018). The 2011 LiDAR and derived pre-disturbance DTMs were differenced into a DOD, and results were summarized with ArcGIS Zonal Statistics.

## 2.4 Catchment scale mapping and analyses of sediment flux

To examine fine-scale patterns of thaw-slump occurrence, network connectivity and change in these parameters through time we investigated the 800 km$^2$ Willow River catchment on Peel Plateau (Fig. 1) because the fluvially-incised stream network straddles the late Wisconsinan glacial limit and much of the basin is intensely affected by thaw slumping (Lacelle et al., 2010). To examine temporal change in this catchment we created digitized inventories of active thaw slumps from the
Willow River catchment utilizing cloud-free Landsat imagery from 1986, 2002 and 2018 described in Rudy and Kokelj (2020). To further explore catchment-scale patterns of slope disturbances over larger areas, and potential contrasts between ice-rich glacigenic study regions we used 2016-2017 Sentinel imagery to re-assess the 2004-2005 slump mapping determined using SPOT 4/5 satellite imagery (Segal et al., 2016c, d) for a 3,520 km$^2$ area of the northern Peel Plateau and 1,220 km$^2$ of southeastern Banks Island (Fig. 1) (Rudy et al., 2020). Catchment areas upstream of thaw slumps were





estimated using CDEM. Tau-DEM (v.5.3) Fill, D8, and Flow Accumulation algorithms (http://hydrology.usu.edu/taudem/)
(Tarboton, 1997) were applied to trace the drainage network and upstream catchment areas.

Sediment fluxes for Peel Plateau streams draining glaciated and unglaciated terrain were calculated to assess effects of
catchment size and thermokarst disturbance (Shakil et al., 2020b). Instantaneous total suspended sediment (TSS) fluxes (mg

$km^{-2} s^{-1}$) were compiled from sampling during the summer flow period (July – September) for 2010, 2015-2017. Sampling
procedures and catchment delineations are elaborated in Shakil et al., 2020a. Historical TSS yields also constrained to July
$1^{st}$ – September $14^{th}$ were obtained for the Peel River at Fort McPherson (Water Survey of Canada, station 10MC002), by
pairing TSS spot sampling with discharge and normalizing to a 70,600 $km^2$ watershed area.

## 2.5 The influence of thaw-driven mass wasting on drainage from continuous permafrost of northwestern Canada

Here we summarize methods to identify individual segments of the hydrological network affected by active thaw-driven
mass wasting and the framework to map the potential propagation pathways of slope thermokarst effects through watersheds
for Arctic drainage from continuous permafrost of northwestern Canada (Fig. 1). Geoprocessing steps are in Supplementary
methods 1-3 and the mapping procedure and raw data are in Kokoszka and Kokelj (2020). In summary, we used
georeferenced, 10m resolution SPOT 4/5 (2004-2010) (NWT Centre for Geomatics, 2013) and Sentinel 2 (2016-2017)

orthomosaics to identify hydrological segments affected by slope thermokarst features for individual stream network, lake,
and coastline segments from the 1:50,000 National Hydro Network (NHN) dataset (Natural Resources Canada, 2016) using
ArcMap 10.6. The Arctic drainage area contains 68 NHN Work Units (subcatchments). The NHN Primary Directed Network
Linear Flow (PDNLF) Shapefiles include primary (main route) stream segments and polylines through lakes, and the NHN
Waterbody and Littoral Shapefiles represent lakes and coastlines, respectively. This dataset was combined to define the

entire NHN hydrological network for the 994,860 $km^2$ drainage area (Figure 1).

A 7.5x7.5 km grid system guided the systematic inventory of slope thermokarst affected hydrological segments in each of
the 68 NHN work units (Kokoszka and Kokelj, 2020). NHN hydrological features were designated as 'directly affected'
where one or more active slope-thermokarst features were in contact with, or exhibited clear drainage towards the

hydrological feature. PDNLF features directly influenced by slope thermokarst were assigned a numeric value of '1'. To
enable propagation of effects from lakes, PDNLF features at the lake outflow within directly affected lakes were also
assigned a numeric value '1'. Unaffected PDNLF features were assigned a numeric value '0' to indicate no thermokarst effect.
All mapping was reviewed for accuracy and consistency.

The potential for downstream propagation of slope thermokarst effects was determined by generating a network for each of
the 68 Work Units using RivEX 10.25 (Hornby, 2017). A river network was constructed for each PDNLF dataset based on
the topology of the original NHN data, where the digitized direction corresponded with flow direction. PDNLF features



directly affected by one or more slope-thermokarst features were assigned a value of '1' as indicated above. The 'Accumulate Attribute Tool' in RivEX 10.25 was used to sum the number of all upstream slope thermokarst affected

segments and tabulated for each Work Unit. One 800 km$^2$ catchment (Willow River, see Fig. 1) representing a subset of the total stream network was subjected to additional analysis since detailed thaw-slump mapping (Rudy and Kokelj, 2020) was available. The total area of mapped thaw slumps and slides for each hydrological segment was summed and accumulated downstream to portray catchment-scale variation in the intensity of disturbance effects (Supplementary Methods 1).

For the broad and fine-scale analyses the accumulated count of impacted stream segments was used to identify the longitudinal trace of thermokarst effects within each Work Unit (Supplementary Methods 2). Stream features with an accumulation value > 0 represented the potential propagation pathway of effects through the network. Propagation of effects across watershed scales involved transferring accumulation values from upstream to downstream Work Units. Hydrological features that were influenced by the downstream propagation of slope-thermokarst effects were identified as indirectly

affected PDNLF features with an accumulation value > 0 and a numeric value of 0. PDNLF features contained within lakes with an accumulation value > 0 indicated indirectly affected lakes, although these lakes could also be attributed as directly affected if the shoreline hosted slope thermokarst features. Coastal segments adjacent to accumulated PDNLF features where streams discharge to the ocean were identified as indirectly affected coastline. After all Work Units were processed, the data were merged to produce a downstream trace with upstream accumulation values for the entire 994,860 km$^2$ drainage area.


To summarize information on the distribution of watershed effects, Strahler Order was computed (Supplementary Methods 3) for the 4 major watershed areas of Banks Island (70,794 km$^2$), Amundsen Gulf (90,288 km$^2$), Peel River (76,506 km$^2$), and Keele/Redstone (39,957 km$^2$) (Fig. 1). For each major watershed, the respective Work Unit and PDNLF Shapefiles were merged and the 'Strahler Order Tool' in RivEx 10.25 was used to compute Strahler Order for each PDNLF polyline. For

each sub-basin, directly and indirectly affected streams and lakes were summarized by Strahler Order.

We summarized the total length of thermokarst-affected stream networks and the coastal margins, as well as the total number of thermokarst affected lakes, including lakes influenced by the downstream propagation of slope-thermokarst effects. Mid-points of the directly affected stream, lake and coastal segments were synthesized in ArcGIS to create a kernel density map

portraying the distribution and density of thaw-driven mass wasting effects on hydrological segments across the study area. The routing of thermokarst effects were portrayed through the downstream trace and accumulation values at the coastline to provide a relative estimate of the magnitude of upstream effects at the point where hydrological networks discharge to the ocean. This current method does not account for variation in transport gradients, catchment sinks, nor intensity of each disturbance, and therefore represents a semi-quantitative, time-transient snapshot of thaw-driven mass wasting effects on

drainage from continuous permafrost of northwestern Canada.



## 2.6 Statistical analyses

The software package SPSS Statistics 21 (IBM) was used for data exploration, deriving summary statistics, testing inferential qualities, and regression analyses for all LiDAR and UAV derived datasets presented in this study. All summary
statistics are provided in tabular format in the body of the manuscript or Supplementary materials. Thaw-slump area, volume and maximum depth of disturbance concavity did not meet assumptions of normality and were logarithmically transformed prior to performing a t-test to assess differences in slump size indices between physiographic regions, and a one-way analysis of variance and a Tukey's HSD post-hoc test to assess differences across categories describing downstream connectivity. Regression analyses between thaw-slump area and disturbance volume, and thaw-slump area and maximum
concavity depth were also performed on the logarithmically transformed datasets. Regression diagnostics included testing the normality of residuals, and Cook's distance to test for the presence of unduly influential points. Additional statistical analyses assessing slope thermokarst conditions in the Willow River catchment and potential changes through time were performed in RStudio (Version 1.1.463) using the "stats" and "FSA" packages (R Core Team, 2017). In this case study, non-parametric statistical testing (Kruskall-Wallis and Dunn's post-hoc tests) was implemented to assess potential differences in
slump size indices between 3 different time periods.

## 3 Results

Here we document: (A) the rapid evolution of thaw-driven mass-wasting and slope-to-stream coupling, (B) the patterns of thermokarst effects within catchments, and (C) their propagation through hydrological networks across watershed scales.
The first three subsections focus on slope-scale processes with emphasis on the thresholds that have transformed connectivity between thawing slopes and downstream systems (3.1, 3.2), and the scaling of thaw-slump dimensions associated with areal enlargement (3.3). The fourth and fifth subsections (3.4, 3.5) focus on quantifying the changing patterns and magnitudes of thermokarst and the potential downstream effects within hydrological networks up to the medium catchment ($10^4$ km$^2$) scale. The final subsection (3.6) provides flow accumulation analysis results to explore the patterns and
to project the routing of slope thermokarst effects through Arctic drainage networks from continuous permafrost of northwestern Canada.

### 3.1 Thaw slump intensification transforms slope-to-stream connectivity

To investigate sediment transfer associated with intensifying slope thermokarst and the evolving linkages between thawing slopes and downslope environments we used UAV-derived survey data from 2018 to quantify thaw slump and debris tongue
volumes for a size and age slump disturbance continuum on the Peel Plateau (Table 1, S1; Fig. 2). The smallest slump (slump D1) which was a decade old, had an area of 2.8 x $10^3$ m$^2$, an eroded volume of 4.6 x $10^3$ m$^3$, and no debris tongue accumulation (Table 1). The second-largest feature (slump CB), which initiated around 2002-2004 had recently accelerated growth, displacing 1.36 x $10^5$ m$^3$ and forming a debris tongue comprising 20% of the scar volume (Figs. 2e, h; Table 1). The geomorphic acceleration of slump CB occurred through our monitoring period and is addressed in Sect. 3.2. Over 2 decades,





two larger slumps, FM3 and Husky, displaced 5.0 x $10^5$ m$^3$ and 6.9 x $10^5$ m$^3$ of material producing debris tongues of 1.5 x

$10^5$ m$^3$ and 3.7 x $10^5$ m$^3$, respectively (Table 1). The valley-fills exceed 15 m thickness and 1 km length (Fig. 2h), comprising

about 30% (FM3) and 50% (Husky) of the estimated scar cavity volumes (Table 1). At Husky slump, the debris tongue

raised the base-level of the master (trunk) stream, elevating and laterally displacing the stream channel against the side of the

valley, causing slope erosion and secondary slump initiation. Enlargement of the Husky slump led to the breaching and rapid

drainage of a lake in 2015, which flushed scar zone materials down valley (Video S1).

**Table 1.** Scar and debris tongue areas and volumes (erosional/depositional) and debris tongue age for a slump-size continuum from September 2018 field observations.

| Slump | Location | Tongue Age | Scar | | Debris Tongue | |
|---|---|---|---|---|---|---|
| | | (yrs.) | Area (m$^2$) | Volume (m$^3$) | Area (m$^2$) | Volume (m$^3$) |
| D1 | 67.1771° N -135.7555° W | n/a | 2,853 | -4,565 | Not present | Not present |
| CB | 67.1814° N -135.7295° W | <5 | 34,084 | -136,496 | 12,007 | 21,567 |
| FM3 | 67.2539° N -135.2732° W | >10 | 73,518 | -501,916 | 31,799 | 147,025 |
| Husky | 67.5207 ° N -135.3005 W | >10 | 64,386 | -690,687 | 73,696 | 377,666 |
| FM2 | 67.2545° N -135.2286° W | >30 | 337,901 | -6,003,927 | 179,000 | 1,949,711 |




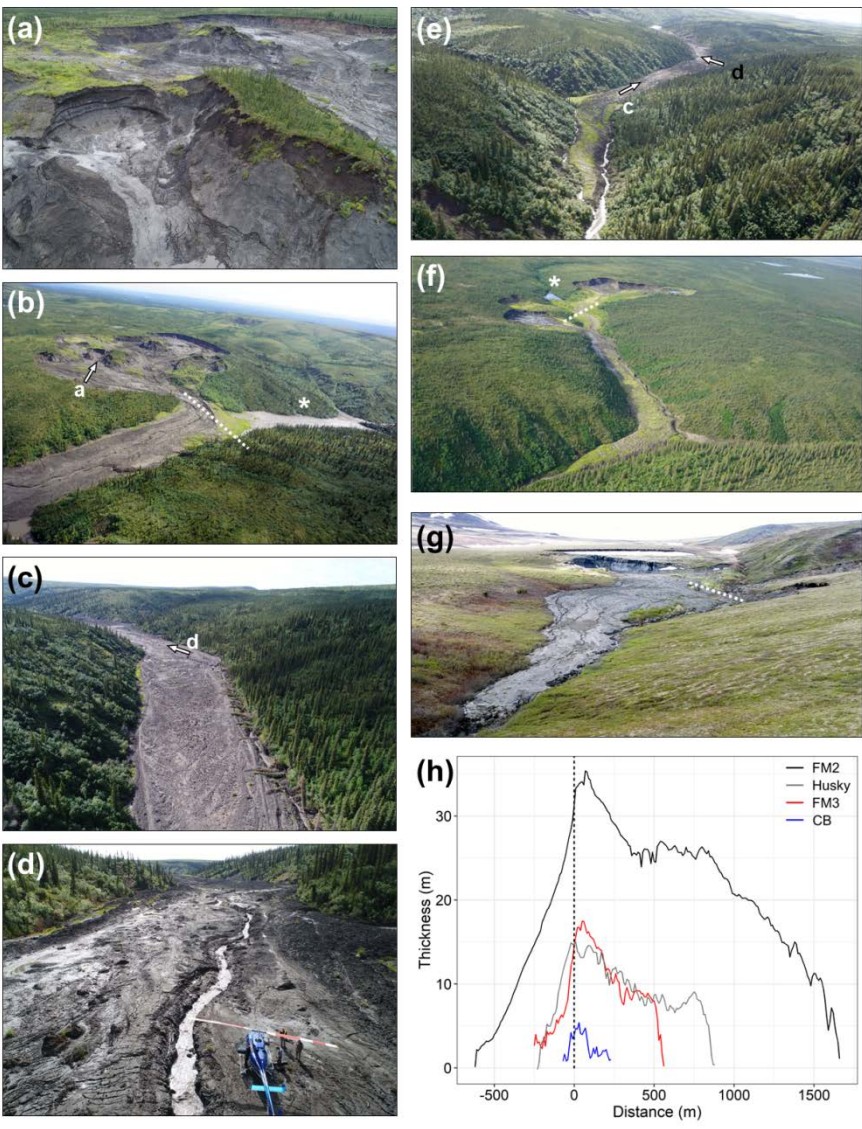

**Figure 2.** Plate showing debris tongue deposits and elevation normalized debris tongue profiles. (a-e) FM2 mega-slump and debris tongue. (a) FM2 headwall retreat showing banded relict massive ice and relatively thaw-stable organic deposit and erosion of glaciofluvial deposit in upper right (background); (b) FM2 scar area and chute (dotted white line) where sediment enters the valley to form the debris tongue and the debris-dam lake (*); (c-e) show FM2 debris tongue from several vantage points indicating the magnitude of the colluvial deposit, initial stages of stream incision (d), and side valley erosion (c, e); (f) Husky slump scar and debris tongue, secondary slumps on left, debris-dam lake (*) and slump-drained lake and residual pond to right of upper slump; (g) Oblique photograph of CB in June 2019 showing debris tongue in foreground, secondary slumps on right side of valley, large polycyclic (secondary) headwall in the central scar zone, and small upper (primary) headwall with snow patch in background. (h) Elevation normalized debris tongue profiles for study slumps; White dashed lines on (b), (f) and (g) indicate 0 m distance on graph (h) where the debris flow enters the downstream channel. Letters and arrows on photographs are reference points indicating location and view of associated photographs.



In 2018, slump FM2 was an order of magnitude larger than the other disturbances, and has been a site of recurrent erosion
for over half of a century (Lacelle et al., 2015). About 30% of the scar cavity volume (6.0 x $10^6$ m$^3$) has been accounted for
in the valley-fill (Table 1; Fig. 2g), consistent with excess ice content estimates of 50 to 70%. Increasing thaw-driven
sediment flows coincided with the regional acceleration of slump activity in the late 1990s (Kokelj et al., 2015), enlarging
the debris tongue to about 2 km length and 35 m thickness by 2018 (Figs. 2 a-f) (van der Sluijs et al., 2018). By filling the
trunk valley, these very large deposits flatten the upstream energy profile, creating depositional environments such as the
debris-dam lake at FM2 (Fig. 2b), which has infilled with sediment, supplied from erosion of the FM3 debris tongue situated
about 1 km upstream. Fluvial incision of the FM2 debris tongue (Fig. 2d) and pinning of the stream channel to the valley
wall (Fig. 2c) is causing high rates of erosion. However, the persistence of these massive deposits demonstrates that thaw-
driven sediment supply exceeds transport capacity of headwater streams by several orders of magnitude.

### 3.2 The thaw-driven evolution of slope-to-stream connectivity

The abrupt transition of small valley-side thaw-slumps into larger, more dynamic failures connected to downstream
environments is transforming the geomorphology of ice-rich glaciated landscapes. These dynamics were captured through
LiDAR and multi-year UAV observations from Peel Plateau (Fig. 3). Thaw slump CB was initiated by stream erosion
around 2002-2004, and gradually back-wasted up an 8 to 12° slope to form a 25,900 m$^2$ disturbance by 2011, comprised of a
shallow scar zone (mean depth = 2.2 m, SD = 2.1 m) with an upper zone of mud slurry and a lower, vegetated accumulation
zone. The headwall, up to 7.0 m high, exposed the active layer underlain by the early Holocene paleoactive layer and
Pleistocene age icy permafrost at depth (Lacelle et al., 2019). In 2015 the scar area had increased to 33,370 m$^2$ (+29%), but
maximum headwall height (5.8 m) and mean concavity depth (1.9 m; SD = 1.9 m) decreased as thawed materials
accumulated in the scar zone (Figs. 3b, d, e). Over a decade of slump growth yielded material redistribution constrained
largely to the slope system. However, a major episode of mass-wasting initiated in summer 2017 evacuated the materials
accumulated within the slump scar zone to form a 2.2 x $10^4$ m$^3$ debris tongue comprised of colluvium transported up to 200
m down valley, and a ~12.5 m high headwall in the central scar zone concavity. This large, lower headwall observed in fall
2018 exposed about 1 metre of colluvium overlying icy sediments that extended to a depth of at least 14 m below the pre-
disturbance surface (Figs. 2g, 3c-e). Gradual retreat of the upper headwall from 2015 to 2018 increased the scar footprint by
only 2%, but evacuation of colluvium in 2017-2018 more than doubled the mean depth and total volume of the scar cavity
(Fig. 3). Thaw-driven, rainfall-enhanced re-mobilization of scar-zone colluvium coupled the slope and stream valley by
placing large volumes of erodible substrate into the channel. Evacuation of materials from the scar zone exposed a large
secondary headwall strengthening feedbacks that drive slump proliferation (Kokelj et al., 2015).


**Figure 3.** Thaw slump growth (slump CB), material displacement and evolution of downstream connectivity 2011 to 2018, Peel Plateau, Northwest Territories Canada. (a) 2011 difference from the reconstructed pre-disturbance terrain surface; (b) 2015 terrain differenced from the 2011 terrain; (c) 2018 terrain differenced from the 2015 terrain, showing polycyclic behaviour including thaw-driven evacuation of scar materials and development of a prominent lower "secondary" headwall. Slump and debris tongue area and volumes are estimated relative to an estimated pre-disturbance surface. (d) The evolution of thaw slump footprint from 2011 to 2018 and the topographic transect in (e) showing cross-sections through the slump and along the valley for 2011, 2015 and 2018.





### 3.3 Quantifying slump enlargement and downstream connectivity across the subarctic-low Arctic transition

Thaw-slump size indices derived from 2011 LiDAR were compiled to investigate relationships between geomorphology, and
downstream connectivity, for the fluvially-incised Peel Plateau and lake-rich Tuktoyaktuk Coastlands and Anderson Plain
(Fig. 1). Within the two LiDAR corridors, scar areas ranged from 242 to 253,900 $m^2$ with a median of 3,323 $m^2$. Volume
displacement ranged from 130 $m^3$ to 3,653,400 $m^3$ with a median of 3,859 $m^3$, and maximum difference between estimated
pre-disturbance surface and disturbed scar surface (referred to as concavity depth) ranged from -1.5 m to -28.0 m with a
median of -3.4 m (Table S2). The accumulation of thawed colluvium on the lower slope of smaller slope-side disturbances
can produce lobes that manifest as positive relief features (Fig. 3a, b). The larger disturbances were characterized by entirely
negative relief relative to the pre-disturbance terrain. The slump size indices are distributed across 3-4 orders of magnitude in
scale, indicating the geomorphic significance of the largest features. Paired t-tests on log-transformed data indicated that the
mean of all slump size indices (area, volume, maximum concavity depth) for the higher relief, fluvially-incised Peel Plateau
were significantly greater (P<0.02) than in the rolling, lake-rich terrain of the Anderson Plain and Tuktoyaktuk Coastlands.


Thaw slump populations were grouped by their connectivity with the downstream environment. One-way ANOVA indicated
significant differences in the means amongst downstream connectivity groups for slump area ($F_{2,\ 68}$=32.6 P<0.001),
concavity volume ($F_{2,68}$=40.9; P<0.001), and maximum depth ($F_{2,\ 68}$= 35.2; P<0.001) (Fig. S1). Tukey's HSD post-hoc test
indicated significant differences amongst all connectivity groups with disturbance area (P<0.02), volume (P<0.005) and
maximum concavity depth (P<0.017) increasing with the strengthening of downstream connectivity. This pattern supports
subsections 3.1 and 3.2 indicating that disturbance enlargement is linked to the coupling of slopes and downstream systems.

To further explore the slope-scale geomorphic implications of slump enlargement we examined the relationships between
slump scar area and volume, and scar area and maximum concavity depth. A linear model fit through the logarithmically-
transformed disturbance area and volume data (Fig. 4a) reveals a power-law relationship:

(log Volume) = -1.44 + 1.42(log Area)                    (Eq. 1)

The model coefficients are comparable with those describing area-volume relationships of landslide populations for a
number of studies from temperate environments across a range of failure mechanisms, material properties and geological
settings (Klar et al., 2011). Figure 4b shows the power-law relationship between scar area and maximum concavity depth
described by:

(log ConcavityDepth) = -0.76 + 0.36 (log Area)                    (Eq. 2)

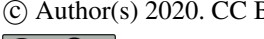



where the depth of permafrost thawed generally increases as thaw slumps enlarge. Scatter in both of these relationships reflects inherent differences in landscape, soil properties, geomorphic setting and the stage of slump development (Figs. 2, 3). However, the relationships indicate that slump area can be used to estimate the volume and thicknesses of permafrost thawed, improving our ability to quantify the role of thermokarst mass wasting in landscape evolution. Further investigations are required to determine whether these relationships will vary with landscape, material properties and failure mechanisms.


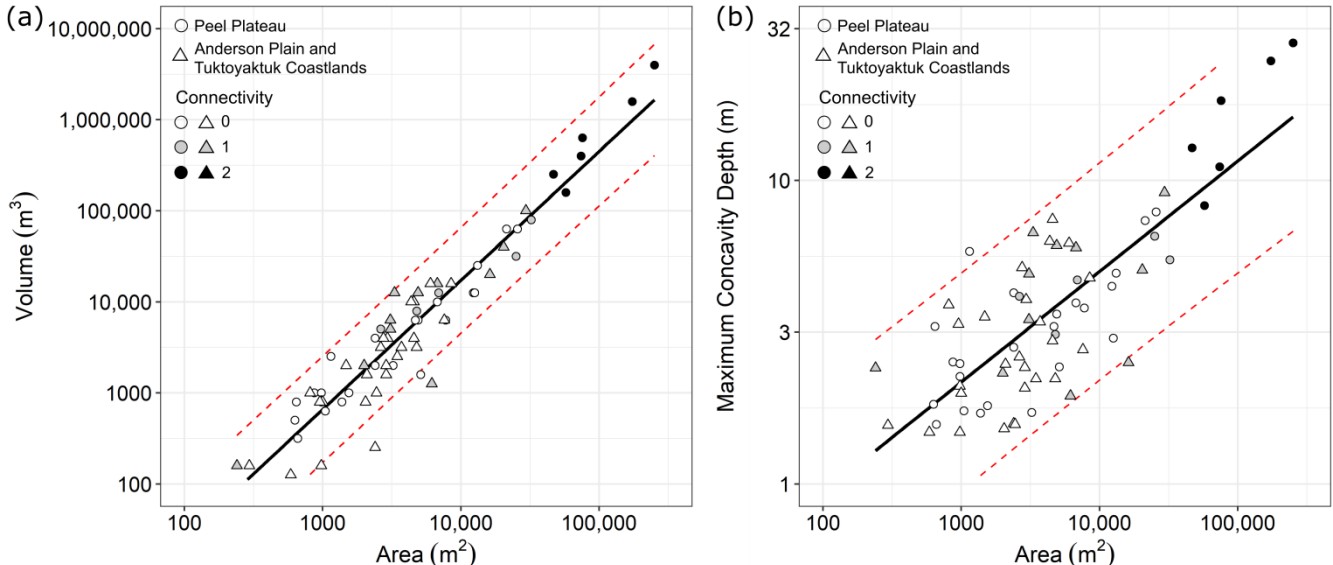

**Figure 4.** Relationships between: (a) slump scar area and volume ($R^2 = 0.90$; F=641.9; N=71; P<0.0001), and (b) slump scar area and maximum depth of concavity ($R^2 = 0.63$; F=116.2; N=71; P<0.0001) for disturbances in the fluvially-incised Peel Plateau, and lake dominated Anderson Plain/Tuktoyaktuk Coastlands. Connectivity between slope and downstream environment is categorized by: 0) no connectivity; 1) physical connection between bare scar and downstream environment; and 2) evidence of downstream deposition of
materials into valley, stream channel or lake. Red dotted lines show 95% confidence limit. Regression diagnostics indicate that residuals are normally distributed and there are no unduly influential points.




### 3.4 Analysis of catchment-scale slope thermokarst effects, Willow River

Thaw-driven mass wasting and slope denudation for 1986-2018 were quantified for the 800 km$^2$ Willow River catchment (Fig. 1). The fluvially-incised stream network drains ice-rich glacial tills and unglaciated scree slopes of the Richardson Mountains into the Mackenzie Delta. Retrogressive thaw slumps are the dominant mode of slope failure; however shallow landslides and few large translational failures also occur (Figs. 6, S2) (Lacelle et al., 2010). Bare or sparsely vegetated thaw-slump scar and debris tongues mapped using Landsat imagery (Rudy and Kokelj, 2020) indicated a fourfold increase in the

number of active thaw slumps from 1986 to 2002, and another 2.5-fold increase by 2018 (Fig. 5a). These results contrast with earlier assessment by Lacelle et al., (2010) in part due to a significantly larger catchment area in this study, different time scales, and nature of imagery utilized. We employed the empirical models developed in Section 3.3, to convert digitized slump scar areas to estimates of slump volume and depth of maximum thaw for the three time periods. Cumulative disturbance area and volume showed 34 and 80 fold increases over the intervals of record (Figs. 5b, c). The median area of

the active thaw slump populations for 1986, 2002 and 2018 was 4,510 m$^2$, 5,680 m$^2$ and 8,290 m$^2$, respectively (Fig. 5d, Table S3). Comparison with mean values for the same time periods of 5,000 m$^2$, 9,770 m$^2$ and 17,910 m$^2$ (Table S3) highlights the increasingly skewed nature of the population distributions as disturbances enlarge (Kokelj et al., 2015). Approximated mean slump volumes and maximum concavity depths of active thaw slumps in 1986 were 6,900 m$^3$, and 3.6 m, increasing to 19,690 m$^3$, and 4.1 m in 2002 and to 59,030 m$^3$, and 5.1 m by 2018 (Table S3). The estimated volumes of

the largest disturbances have increased by two orders of magnitude, and there has been at least a three-fold increase in the approximated maximum concavity depth (Fig. 5), although the models appear to provide conservative high end estimates of both parameters (Fig. 4). Kruskall-Wallis rank sum tests indicated significant differences in slump size indices amongst the different time periods ($\chi^2(2)$ = 22.825; P<0.0001). Dunn's post-hoc testing showed no significant increases in disturbance area, volume or maximum concavity depth from 1986 to 2002, however increases in the population medians were significant

for all size-related indices from the first two time periods to 2018 (Fig. 5d-f; Table S4), suggesting a period of slump initiation followed by accelerated enlargement from 2002-2018. During the 2002-2018 period, development of numerous debris-tongue deposits indicates major strengthening of connectivity between slopes and streams (Fig. 5a). The catchment-wide cumulative slump volume estimates of 0.15 x 10$^6$ m$^3$ in 1986, 1.43 x 10$^6$ m$^3$ in 2002 and 11.70 x 10$^6$ m$^3$ in 2018 reveal a non-linear increase in thaw-driven geomorphic activity across two orders of magnitude over three decades (Fig. 5c).

Normalizing by catchment area and differencing with the preceding time interval, the thaw slump component of surface lowering amounts to 0.1 mm yr$^{-1}$ for 1986-2002 and 0.8 mm yr$^{-1}$ for 2002-2018.  In a relatively quiescent landscape that is frozen for almost two-thirds of the year, this represents a dramatic change in sediment delivery regime from thawing slopes to the stream systems.

The major intensification of thaw-driven mass wasting in the Willow River catchment (Fig. 5) predominantly occurs on the uppermost slopes in the drainage networks, mainly affecting first and second-order streams (Fig. 6). The exposure of several



kilometres of slump headwall confirms that the low-order stream networks are incising ice-rich glaciated terrain and that the majority of materials mobilized by the largest disturbances comprise Pleistocene sediments and relict ground ice (Fig. 6; Video S2). Steeply-incised valley sides of fourth and fifth-order channel segments which occupy the broader valley of Willow River main stem are also affected by a number of large, deep-seated translational failure complexes (Figs. 6, S2) that exhibit bedrock control, as well as thaw-driven flows. The incised valleys are susceptible to shallow landslides, and although they transfer comparatively modest amounts of sediment to the stream network, they contribute to thaw slump initiation by exposing ground ice on steep slopes. Our mapping of the Willow River catchment indicates that 116 km out of 861 km of stream segment length was directly affected by slope thermokarst, with downstream accumulation increasing the length of the affected network by 246 km, to 42% of the entire stream network. The downstream translation of thaw-driven slope sediment mobilization (Figs. 5, 6) through the Willow River fluvial network is indicated by the rerouting of the outflow channel to the Mackenzie Delta in 2007-2008, leading to the rapid infilling of a 3.4 km$^2$ lake over the span of a decade (Fig. 6c; Video S3).



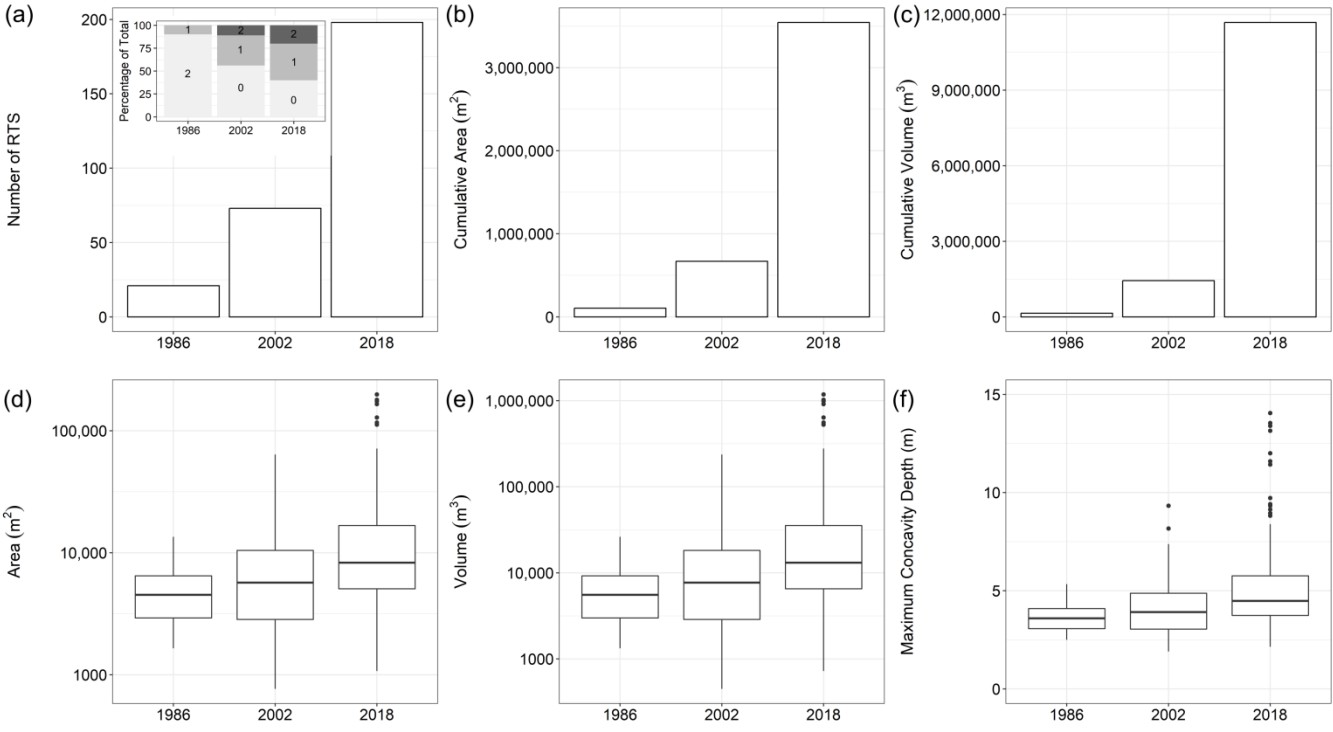

**Figure 5.** Thaw-driven geomorphic disturbances in the Willow River catchment for 1986, 2002 and 2018. Bar graphs show: (a) Slump
counts with inset showing proportion of retrogressive thaw slumps (RTS) with downstream connectivity, (b) cumulative slump area, and
(c) estimated cumulative slump volume. Box and whisker plots show; (d) thaw slump area, (e) approximated volumes, and (f) maximum
concavity thicknesses.  Note logarithmic scale on (d) and (e). On Plot (a) inset, the connectivity between the slope and downstream
environments is categorized by 0) no connectivity, 1) physical connection between bare scar area and downstream environment, and 2)
evidence of downstream deposition of materials into valley, stream channel or lake. Thaw slump volume and maximum concavity
thickness estimated using relationships shown in Figure 4. The spatial dataset is available in Rudy and Kokelj, 2020a.



**Figure 6.** Thaw-driven landslide impacts to the fluvial network of the Willow River, Mackenzie Delta area. Red lines along the drainage network highlight impacted stream segments, weighted by cumulative slope thermokarst area (ha), and the downstream accumulation of impacts through the fluvial system. The black proportional circles indicate the total landslide scar area >1ha per affected stream segment, including thaw slump, and shallow and deep translational slides. (a) The area weighted distribution of thaw-driven landslide impacts by Strahler order within the Willow River fluvial network. (b) Landsat images (1986-07-07) and (2019-07-27) indicating increases in thaw-driven landslide erosion developed since the late 1990s, and corresponding oblique aerial photographs of the largest thaw slump within the Landsat plate. (c) Landsat images of rapid lake infilling and delta development where Willow River rerouted in 2007-2008 into Willow Lake (outlined in Orange), Mackenzie Delta, and oblique photograph showing the new alluvial deposit. The abandoned channel is shown in dark blue. Base map is from ESRI ArcGIS Online.





### 3.5 Catchment-scale patterns of thaw slump effects and sediment flux across catchment scales

The patterns of slope thermokarst distribution and intensification were assessed across catchment scales for a 3,520 km$^2$ area on Peel Plateau and contrasted with a 1,220 km$^2$ area on southeastern Banks Island (Fig. 1). The distribution of slope disturbance determined from SPOT 4/5 2004-2005 and 2018 Sentinel imagery on Peel Plateau shows increasing frequency across all size classes (Fig. 7a). This pattern reflects initiation of new disturbances including some smaller slides, as well as perpetuation and enlargement of existing features. On southeastern Banks Island, overall disturbance density was greater, but the relative increase in disturbance incidences over the past decade was lower than for Peel Plateau. A distinct shift towards larger features on Banks Island suggests disturbance growth and coalescence (Fig. 7b). Figure 7c provides a cumulative summary of the points at which the river network is being impacted by major thaw-driven mass wasting activity. Evidently major disturbances tend to impact the channel at 1-100 km$^2$ catchment scales, consistent with patterns observed for Willow River (Fig. 6a). The temporal shift towards direct effects to larger channels in the Peel Plateau region reflects the greater watershed sizes, and thus more available downstream terrain for disturbance proliferation (Fig. 7c). Some of the main Peel River tributaries are increasingly being affected by large slope thermokarst failures and shallow slides. On Banks Island, the smaller fluvial networks were already highly impacted by thaw slumping in 2005 (Segal et al., 2016a, Lewkowicz and Way, 2019), leaving less available terrain for expansion of network impacts, as indicated by the slightly more subdued increases in slump incidence, and a static pattern in distribution of catchment disturbances between the two time periods (Figs. 7b, c). Figure 7d reinforces that the geomorphic and geochemical impacts to headwater streams tend to be more intense than direct impacts to larger systems, because the size of the disturbance relative to the watershed area will be far greater. Further downstream, the overall impacts from individual disturbances and cumulative effects will be proportionately less, given the much larger upstream catchment area (i.e. trunk channel discharge) and greater stream transport capacity, relative to sedimentary or geochemical yields from the disturbance.

There is a notable dearth of river sediment monitoring over the vast, thermokarst sensitive region of northwestern Canada. A compilation of available grab samples of stream sediment concentrations coinciding with discharge measurements (Shakil et al., 2020b) within and adjacent to Peel River basin for catchments from 10$^{-1}$ to 10$^4$ km$^2$ indicate a negative trend in specific low-flow fluxes with catchment area, consistent with observations of suspended sediment concentrations in glaciated permafrost terrain reported by Kokelj et al., (2017a). Estimates of low-flow sediment fluxes for small thaw-slump affected catchments are up to two orders of magnitude greater than for larger watersheds, and several orders of magnitude greater than for undisturbed and unglaciated catchments (Fig. 7e).




**Figure 7.** The patterns of slump intensification in fluvial networks for 2004-2005 and 2017 in Peel Plateau (3,520 km$^2$) and eastern Banks Island (1,220 km$^2$) regions. (a, b). Size distribution of thaw slumps for the two time periods where white indicates 2004-2005, black indicates 2018, and grey indicates overlap between 2004-05 and 2018 distributions; (c) Cumulative thaw-slump disturbance count plotted against catchment area; (d) The ratio of disturbance area to upstream catchment area plotted against upstream catchment area; and (e) July-September instantaneous sediment flux rates for streams and rivers the Peel watershed and adjacent areas.



### 3.6 Slope thermokarst effects propagate through hydrological networks to the ocean

Here we project the potential cascade of slope thermokarst effects through Arctic drainage networks from continuous permafrost of northwestern Canada (Fig. 8). The summary of active slope thermokarst by Strahler stream order for Banks

Island (70,794 km$^2$), Amundsen Gulf (90,288 km$^2$), Peel River (76,506 km$^2$), and Keele/Redstone (39,957 km$^2$) watersheds indicates that the greatest abundance and highest density of directly affected thermokarst stream segments was on Banks Island (11.8/1000 km$^2$) and the lowest was in the Amundsen Gulf (2/1000 km$^2$). However, major hotspots associated with ice-rich moraine, glaciofluvial and glaciolacustine materials occur within all four watersheds where disturbance densities are an order of magnitude greater than the watershed average. About 71% of direct slope thermokarst effects to streams, and

over 81% of direct effects to lakes occur within first and second order hydrological segments (Fig. 8a), corroborating catchment-scale patterns (Figs. 6, 7), and indicating that thousands of small streams and hundreds of lakes across northwestern Canada are directly affected by slope thermokarst (Fig. 8b). The major direct effects on hydrological networks are concentrated in well-defined geographical areas, with large regions free of major thaw-driven mass-wasting. For Banks Island and the Peel and Keele-Redstone watersheds, low disturbance densities occur west of the late Wisconsinan ice-limit.

Slope thermokarst was sparse in unglaciated Cordilleran areas and glaciated Precambrian Shield with patchy till veneer. Low disturbance density also characterizes the Taiga Plains of central Mackenzie Valley, however some valleys incising tills and glaciolacustrine deposits are subject to thaw-related failures, including the Mackenzie River main stem (Figs. 8, S3iii).

As a broad summary, slope thermokarst activity directly affects over 6,750 km of stream segment length, 1,370 lakes and

about 900 km of coastline in the continuous permafrost drainage area of northwestern Canada (Fig. 8). About 48,800 km of stream network and over 5,600 lakes are susceptible to downstream or "indirect" effects, reflecting 7.2- (streams) and 4.1- (lakes) fold increases over the extent of direct hydrological effects. The much greater extents of downstream effects (Fig. 8a) occur because slope thermokarst predominantly impacts low-order streams often in headwaters of the hydrological network. A high density of slope thermokarst features connected through small fluvial systems drain eastern Banks Island and

northwestern Victoria Island indicating major potential to propagate sedimentary and geochemical effects directly to the coastal zones of the Prince of Wales Strait (Figs. 7c, 8). Several Arctic drainages northeast of the Mackenzie Basin with areas of high slope thermokarst density have the potential to propagate effects. Ice-rich permafrost characterizing a diversity of ice-marginal glacigenic deposits of the Anderson Plain and Tuktoyaktuk Coastlands (Fig. 8ii) surround the Husky Lakes estuary, which integrates the effects of shoreline slumping, discharge from numerous small slump-affected, lake-dominated

catchments, and a few slump-affected northward-flowing rivers, indicating a regional hotspot of accumulated effects (Figs. 8, S3). Coastal erosion along the Yukon North Slope and uplands east of Mackenzie Delta, in conjunction with lake-side disturbances drive the majority of slope thermokarst processes there, whereas fluvial transfer from inland areas to the coast is relatively limited. The highest density of slope thermokarst affects numerous high-energy, fluvial-incised networks draining ice-rich glaciated landscapes of the lower Mackenzie and Peel River basins and these river systems propagate the greatest

magnitudes of sedimentary and geochemical effects to the Mackenzie Delta and Beaufort Sea (Fig. 8).







**Figure 8.** Thaw-driven landslide density and downstream accumulation of effects, western Canadian Arctic drainage from continuous permafrost terrain. Heat map depicts all directly affected stream, lake and coastal hydrological features mapped. All upstream accumulation values mapped within the fluvial network are >2 and at the coast are >9. Accumulated effect contributions of the Mackenzie and Peel Rivers to the Beaufort Sea are routed separately for comparison. (a) Counts of direct and accumulated thaw-driven mass wasting effects to fluvial systems by Strahler order for focal watersheds outlined in blue. (b) Table showing lengths of directly affected hydrological network and accumulated effects, count of directly and indirectly affected lakes and total directly affected coastline for western Arctic drainage from continuous permafrost. Remote sensing examples of thaw-driven downstream sedimentation provided in Fig. S3 for: i) Sachs River and Fish Lake, ii) Miner River inflow to the Husky Lakes estuary and iii) massive deep-seated permafrost failure on Johnson River. Late glacial limit is from Dyke and Prest, 1987, bedrock geology is from Fulton, 1995, and the permafrost boundary is from Brown et al., 1997. Base map is from ESRI ArcGIS Online.



## 4 Discussion


### 4.1 Thresholds and connectivity between slopes and downstream environments

Results indicate that the intensification of thaw-driven mass wasting is altering the Holocene-scale regime of sediment production, mobilization, and delivery in the periglacial landscapes of northwestern Canada (Figs. 2, 3, 6, 7e). Rapid evolution of hillslope-channel coupling has occurred because slope thermokarst disturbances have emerged as multi-decadal,

thaw-driven conveyors of fine-grained materials to the downstream environment. Slope and channel morphology and sediment yields in catchments unaffected by active mass-wasting, and absence of relict valley fills (Figs. 2, 3, 7e) suggests that over the past millennia, the majority of slope failures have restricted material redistribution to the slope system (Figs. 3a, b). Exceedance of critical thresholds controlling sediment detachment from hillslopes driven by interdependent factors of warming, precipitation, permafrost thaw and soil saturation (Lewkowicz and Way, 2019; Segal et al., 2016a; Ward-Jones et

al., 2019) have instigated processes and feedbacks (Fig. 2) (Kokelj et al., 2015), which rapidly mobilize materials and couple slopes with hydrological systems across a range of glacially-conditioned permafrost environments (Fig. 8). The non-linear growth of disturbances over time, strengthening of downstream connectivity (Figs. 2-4), and predominant impacts to low-order streams (Figs. 7c-d, 8a) emphasizes the disproportionate effect that increasing disturbance size is having on periglacial slope evolution and the magnitude and duration of downstream effects.


Recent thaw-driven mobilization of slope materials has increased sediment stores in valley bottoms by several orders of magnitude causing the rapid aggradation of channel beds (Figs. 2, 3), and together with the steepening and lateral displacement of channels, will amplify fluvial sediment transport for decades to centuries. Slope-to-channel connectivity and downstream coupling is particularly enhanced in steeply incised and confined valleys, so that environments like the Peel

Plateau have emerged as hotspots of geomorphic change and downstream effects (Figs. 6, 8) (Kokelj et al., 2013; Zolkos et al., 2018). The transition from the long-term 'supply-limited' norm of the Holocene permafrost geomorphic systems (*sensu* Carson and Kirkby, 1972; Howard, 1994), is striking. The change is emphasized by the unprecedented geomorphic response of long-frozen permafrost slopes, rapidly overloading a network of newly "under fit" streams (Figs. 2, 3, 6, 7e). Enhanced slope-channel coupling and increased network connectivity are evident across the study region (Figs. 6; S3). However, the

downstream conveyance of sediment and geochemical constituents will vary (Kokelj et al., 2013; Shakil et al., 2020b), because the connectivity of slope and hydrological systems are inherently sensitive to substrate properties and behaviour of geochemical constituents, network configuration, stream power, physiography, permafrost conditions and climate drivers including air temperatures and precipitation regimes. While it is difficult to translate thousands of discrete and dynamic slope disturbances into coherent estimates of river impacts at the basin-scale, this has emerged as a central challenge facing

periglacial geomorphologists, biogeochemists and aquatic ecologists in the Anthropocene.



## 4.2 Non-linear disturbance trajectories

At the slope-scale, our nested study highlights the rate and volumes of material mobilised from slope thermokarst
disturbance in glacially-conditioned permafrost terrain (Kokelj et al., 2017a). The thickness of permafrost thawed is
associated with disturbance volume, and both parameters are related to disturbance area via power-law relationships (Fig. 4).
The area-volume model we have derived has coefficients closely comparable to landslide populations in temperate
environments (Klar et al., 2011). The nature of these relationships are complex because permafrost, geomorphic, and
climatic factors can influence the growth trajectory of individual disturbances (Figs. 2, 3), to produce dynamic landforms
over terrain with varied material and topographic properties. Regardless, the relationships determined in this study provide a
first empirical basis for estimating thaw-driven denudation as a function of disturbance area (Figs. 4, 5, 7a), and better
approximation of the geomorphic, sedimentary, geochemical and carbon consequences of intensifying disturbance regimes.
Further investigations are required to determine the potential variability in area-volume relationships with landscape type,
material properties and modes of failure.


The rapid increases in the area of thaw-driven slope disturbance are having a significant influence on terrain evolution, and
downslope sediment and geochemical mobilization. The increasing prominence of larger, and thus proportionately deeper
disturbances, indicates greater volumetric thawing of ice-rich permafrost and mobilization of sediment than would be
anticipated for the equivalent, combined surface area of numerous smaller slumps (Fig. 4; Table 1, 2). It also suggests that a
greater range of material types are likely to be thawed and entrained. In glacigenic deposits, material excavated from deeper
in the soil stratum tend to have lower organic matter contents and is unlikely to have undergone the thaw-induced
geochemical changes that characterise the active layer and near-surface permafrost (Lacelle et al., 2019). Enlargement and
accelerated back wasting of the slump headwall, coalescence of thaw slumps, and polycyclic behaviour (Figs. 2, 3) increase
the production of saturated substrate and the probability of material evacuation from the scar zone. As the magnitude of
thaw-driven slides increases there is also greater potential for underlying bedrock, or in southern permafrost regions,
unfrozen materials, to be exposed, further expanding the trajectories of geomorphic and geochemical change, and potentially
linking surface and subsurface flow paths (Walvoord and Kurylyk, 2016).

The intensification of thaw-driven mass wasting is inextricably linked to the strengthening of lateral and downstream
connectivity (Figs. 2, 3, 4; Tables 2, 3).  For example, as retrogressive thaw-slumps enlarge, greater volumes of material,
varying laterally and stratigraphically are being mobilized, mixed as a saturated slurry, and transported by a suite of mass-
wasting processes to form colluvial deposits that can veneer slopes, accumulate in downstream environments or enter larger
river channels, as well as lacustrine or coastal environments (Figs. 2, 3) (Kokelj et al., 2015; Houben et al., 2016; Ramage et
al., 2018). Thaw-driven debris tongue development in small streams is also commonly blocking drainage and causing debris
dam lakes and ponds to form at hundreds of locations. The rapid intensification of thaw-driven mass wasting combined with





variation in the type of landscapes being impacted, and the nature and magnitude of material mobilized, has a range of geomorphic, biogeochemical and ecosystem implications that require study (Tank et al., 2020; Vonk et al., 2019) so that the broader-scale consequences of accelerating slope thermokarst can be predicted in an informed manner.

### 4.3 The river network: transfer of thermokarst erosional materials from source to sink

Our spatially nested study design provides several insights on the evolving nature of slope-thermokarst effects on hydrological networks. Firstly, acceleration of thaw-driven mass wasting is affecting thousands of first and second-order lakes and streams in permafrost preserved, glacially-conditioned landscapes (Fig. 8). Major sedimentary, geochemical and ecosystem impacts upon low-order stream networks have been abrupt (Kokelj et al., 2013) because the increasing magnitude of disturbance and enhanced downstream connectivity facilitate the cascade of downstream effects through small systems with an inherently limited capacity to absorb disturbance (Figs. 7c, d). Specific examples were provided where slope thermokarst effects to low-order streams can entirely change the character of the stream reach (Figs. 2, 3), the sediment and geochemical fluxes (Fig. 7e) (Kokelj et al., 2013; Shakil et al., 2020a), and the associated habitat potential (Chin et al., 2016). Secondly, thaw-driven sediment and geochemical mobilization is directly impacting, or accumulating to affect thousands of low-order lakes in glacially-conditioned environments (Fig. 8) that are also inherently predisposed to abrupt limnological responses (Houben et al., 2016; Kokelj et al., 2009). Thirdly, the direct effects of slope thermokarst occur predominantly on hydrological systems at the $10^{-1}$ to $10^{2}$ km$^2$ scale (Figs. 5-8). Through tracking changes in the size and distribution of thaw slump features within the Willow River study catchment, we estimated a two order-of-magnitude increase in thaw-driven mass wasting from 1986 to 2018 (Figs. 5a-c), and, integrating sediment and ice loss, about an order of magnitude increase in the contribution to catchment-scale denudation of slopes, from 0.1 mm yr$^{-1}$ (1986-2002) to 0.8 mm yr$^{-1}$ for the 2002-2018 period (Figs. 5, 6). Hillslopes along the margins of 4th and 5th order drainages have also shown increasing responsiveness to thaw-driven instability and lateral river erosion, resulting in direct sediment delivery to the larger streams. Fourthly, fluvial networks integrate a time-transient cascade of upstream effects indicating that water quality, aquatic ecosystems, and fish habitat will be affected across increasingly larger river systems and coastal environments in the study domain (Fig. 8). Unequivocal evidence that thermokarst-derived sediments in addition to geochemical effects (Kokelj et al., 2013; Tank et al 2016) are propagating through fluvial networks is provided by channel re-routing and rapid infilling of a large lake where the Willow River fluvial network discharges to Mackenzie Delta (Fig. 6; Video S3), and by the increases in turbidity of numerous downstream lake, delta, estuary, and coastal environments (Figs. S2i-iii), which coincide with our predicted areas of downstream accumulated effects (Fig. 8).

### 4.4 Hydrological connectivity defines a significant global change hotspot

The significance of thermokarst processes are magnified by the propagation of sedimentary, geochemical and ecological effects to freshwater systems and the marine environment (Ramage et al., 2018; Rudy et al., 2017a; Vonk et al., 2019). While we have not simulated dynamic routing of sediment across the 1,000,000 km$^2$ study area, our nested design



demonstrates evolving linkages between slope thermokarst and hydrological systems (Figs. 2-7), and projects the potential downstream effects at the $10^6$ km$^2$ basin scale (Fig. 8). Rapid proliferation of numerous, major point source disturbances amplifies the potential of downstream cumulative effects as mobilized sediments, solutes, and carbon transit a sequence of storage reservoirs, and reinforce a thermokarst signal that is projected to cascade across increasing watershed scales to marine environments through the coming century. Although this broad-scale analysis (Fig. 8) neglects relative disturbance

intensity, geomorphic and biogeochemical sinks, variations in slope-downstream connectivity, and transport gradients, our projection provides a spatially explicit framework for exploring the distribution and time-transient propagation of slope thermokarst effects. Identification of major thaw-driven sediment and solute source areas, and the primary hydrological networks that convey slope thermokarst effects to coastal environments, emphasizes the interconnected nature of a global thermokarst hotspot (Fig. 8), and could guide future scientific investigations in the study region.


**4.5 Glacial legacy controls intensity of slope thermokarst and thaw-driven reconfiguration of fluvial networks**

Glacial legacy, geomorphic setting, and climate history all combine to determine the distribution and intensity of thaw-driven mass wasting, and the nature and routing of downstream effects across continuous permafrost of northwestern Canada (Figs. 1, 8) (Kokelj et al., 2017a). A cooling trend through the Holocene (Porter et al., 2019) has preserved ground ice and

maintained moraine, glaciofluvial, glaciolacustrine, and glaciomarine deposits in a quasi-stable state. Climate-driven episodes of permafrost thaw, most notably in the early Holocene, have left their imprint in the form of ancient landslide scars, thaw lakes, colluvial deposits, and a regional thaw-unconformity (Burn, 1997; Lacelle et al., 2019; Mann et al. 2010; Murton, 2001). However, millennia of cold-climate constraints on slope evolution and fluvial network development are made apparent by the abundance of steeply incised, or solifluction-smoothed valley slopes underlain by ice-rich permafrost,

and now juxtaposed with adjacent, enormous climate-driven mass wasting features that are mobilizing discordant volumes of materials to low-order hydrological systems that exceed stream transport capacity (Figs. 2, 3, 5, 6) (Kokelj et al., 2015). In continuous permafrost regions, climate-driven rejuvenation of post-glacial permafrost slope evolution is driven primarily by top-down thawing causing shallow sliding and thaw slump development (Figs. 1, 2, 5, 6) (Lewkowicz and Way, 2019; Segal et al., 2016a; Ward-Jones et al., 2019). However, ground warming and a loss of soil strength at depth have also led to

increasing frequency and magnitude of deep-seated rotational and translational slope failures in low and subarctic environments (Fig. S3iii) (Aylsworth et al., 2000). The intensification of precipitation regimes throughout parts of the study area has increased thermoerosion, shallow landsliding, and downslope sediment transfer in active thaw slumps so that thawing slopes are increasingly pushed toward an unstable state (Fig. 2). In the subarctic, ecosystem disturbances such as fire compound effects of climate warming (Holloway et al., 2020), triggering an array of slope mass-wasting processes in

areas already predisposed to slope instability (Fig. 8).

The most intensive slope thermokarst activity in North America, and perhaps across the circumpolar Arctic, has been associated with western margins and recessional positions of the Laurentide ice-sheet (Figs. 1, 8) (Kokelj et al., 2017a)





where preservation of relict segregated and glacier ice (Lacelle et al., 2010; Lakeman and England, 2012; Mackay, 1971; Murton et al., 2005) and aggradation of Holocene ground ice (Burn, 1997; Holland et al., 2020) is hosted within thick glacigenic deposits. These ice-rich glacial materials are derived largely from Paleozoic and Cretaceous sedimentary rocks, and the majority of the deposits include fine-grained, solute-rich materials, easily mobilized by thaw-driven slope failure (Figs. 2, 3) and fluvial processes (Figs. 6, S3) (Kokelj et al., 2013, 2015; Malone et al., 2013). These environments stand in contrast to the relatively stable slopes of glaciated terrain in shield dominated landscapes including the southeastern portions of this study region. Areas of uplifted and incised glaciomarine deposits also fit within the framework of permafrost preserved glacially-conditioned landscapes, and although many are outside of our study domain, significant terrain, sedimentary and geochemical responses to thawing slopes have been documented there (Kokelj and Lewkowicz, 1999; Ward-Jones et al., 2019).

The abrupt intensification of slope thermokarst and the patterns of impacts across fluvial networks of northwestern Canada indicate a deglaciation-phase response pattern (Figs. 2, 3, 6, 8), which reflects a typically rapid period of geomorphic transition (Ballantyne, 2002), that in permafrost regions has experienced a long-term climatic hiatus through a cooling Holocene. While more data are needed to fill in the full spectrum of annual fluvial sediment yield across catchment scales in thermokarst-affected landscapes, there is a consistent pattern of much higher fluxes in headwater systems, the site of 'primary' glacial sediment stores (*sensu* Ballantyne, 2002). This pattern in thermokarst affected permafrost regions contrasts with that found in temperate British Columbia's post-glacial river systems, where primary glacigenic sediment stores have been exhausted: those materials have cascaded through fluvial networks over millennia and are now reworked and transported by larger rivers (>1,000 km$^2$; Church and Slaymaker, 1989). In glacially-conditioned permafrost landscapes, climate-driven mobilization of these glacial sediment stores is currently overwhelming the transport capacity of low-order streams, leading to valley-filling, upstream pond and lake formation, and river aggradation that will reinforce this early-stage paraglacial signal for centuries to come. The prospect that climate-change is renewing the sequence of post-glacial landscape evolution points to the massive potential for thermokarst-driven geomorphic change on ice-rich slopes and the centennial to millennial scale perturbation of downstream fluvial systems (i.e., Church and Slaymaker, 1989), and the receiving lacustrine, deltaic and coastal environments in several Arctic regions (Kokelj et al., 2017a).

## 5. Conclusions

Non-linear intensification of thaw-driven mass wasting is transforming permafrost preserved glacigenic landscapes and downstream connectivity, triggering a cascade of effects that are propagating through Arctic hydrological networks. In the most intensely impacted watersheds, hundred-fold increases in thaw-driven geomorphic activity have been quantified over a three-decade period. Power-law relationships between thaw slump area and volume emphasize the non-linear influence of increasing disturbance area on landscape morphology, slope to stream connectivity and downstream effects. The disequilibrium imposed by thaw-driven release of massive sediment stores from the headwater slopes (1$^{st}$ and 2$^{nd}$-order



streams) reflects a tipping point within these long-frozen permafrost land systems and signals a persistent perturbation of downstream hydrological networks. We can anticipate that thaw-driven mobilization of sediments and soluble materials and their cascade through western Canada's Arctic river networks will enhance connectivity, and fundamentally alter the sediment transporting characteristics of long-quiescent slopes and streams with major implications to downstream geomorphic, geochemical and ecological conditions, and compounding localized thermokarst effects to lake and coastal systems.

We estimate that thaw-driven mass wasting directly affects approximately 6760 km of stream segments, 1380 lakes, and 900 km of coastline across the 1,000,000 km$^2$ Arctic drainage area from continuous permafrost of northwestern Canada. The propagation of thermokarst effects through fluvial networks has the potential to increase the number of affected lakes by a factor of 4.0, and the length of the impacted fluvial network by a factor of 7.0. Projection of accumulated thermokarst effects indicated numerous susceptible lake, delta, estuary and coastal environments throughout the study basin. Major thaw-driven geomorphic activity is concentrated along ice-rich moraine systems that define coastal regions of Banks and Victoria Island bounding Prince of Wales Strait where numerous, small, intensely affected fluvial systems transfer thermokarst-derived sediments and solutes short distances to the coastal environment. In contrast, the thaw-driven sediment and solute loads conveyed by hundreds of upland streams draining glaciated, ice-marginal landscapes of the lower Peel and Mackenzie watersheds are routed into the two major rivers and indicate a long-term cascade of thermokarst effects to the Mackenzie Delta and the Beaufort Sea. The varying intensity of thaw-driven slope disturbances and the propagation of sedimentary and biogeochemical effects (i.e., dissolved vs. particulate; carbon vs. nutrients vs. major ions) across hydrological networks of varying configuration (e.g., channel slope, lake density, size) will define the evolution of downstream geomorphic, ecological and biogeochemical systems over the coming centuries.

Glacially-conditioned permafrost terrain exhibits massive potential for thaw-driven geomorphic transformation, and underlines exceptionally strong climate-geomorphic linkages. Intensifying slope thermokarst effects are propagated through hydrological networks greatly amplifying the extent, magnitude and duration of impacts related to permafrost thaw. This study highlights the interconnected nature of climate-driven thermokarst by quantifying relationships between intensification of disturbance and strengthening of downstream linkages, and by projecting the cascade of effects across hydrological networks to coastal systems of northwestern Canada. Through placing thaw-driven geomorphic phenomena in a geological and hydrological framework, spatial patterns and trajectories of landscape change, as well as environmental consequences can be contextualized and predicted in an informed manner. Glacial legacy, rejuvenation of post-glacial permafrost landscape evolution, and patterns of continental drainage dictate that western Arctic Canada will be an interconnected thermokarst hotspot of global significance through the coming millennia.

*Data availability.* The datasets are referred to in text and links are in the reference list. Framework and geoprocessing steps for developing flow accumulation analysis are available in the Supplementary materials.

*Supplementary materials.* The supplement related to this article is available online at:

*Author contributions.* SVK and JT developed the paper concept. All authors contributed to field data collection and analysis of data and refinement of paper scope. JVS processed and analysed the UAV and LiDAR data with input from SVK and JT. JK and SVK developed the flow network analysis with input from JT and AR. AR, JVS and JK produced final Figures. SVK drafted and revised the paper with input from all authors.

*Competing interests.* The authors declare that they have no conflict of interest.

*Acknowledgements.* The work was supported by the Department of Environment and Natural Resources Climate Change and Northwest Territories Cumulative Impact Monitoring Program of the Government of the Northwest Territories, the Natural Science and Engineering Research Council of Canada, and the Polar Continental Shelf Program, Natural Resources Canada. Long-term support from the Tetl'it and Ehdiitat Renewable Resource Councils, the Inuvik, Tuktoyaktuk and Sachs Harbour Hunters and Trappers Committees, the Inuvialuit Joint Secretariat, the Inuvialuit Land Administration and from the Aurora Research Institute and NWT Centre for Geomatics are gratefully acknowledged. Field support from Christine Firth, Eugene Pascale, Steven Tetlitchi, Alice Wilson, and Billy Wilson are gratefully acknowledged. Photographs in Figure 3 were provided courtesy of Rob Fraser, Canada Centre for Remote Sensing. This research has benefited from discussions with Chris Burn, Douglas Esagok, Rob Fraser, Denis Lacelle, Trevor Lantz, Peter Morse and Stephen Wolfe. Comments by Stephan Gruber have improved the manuscript.

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
