# Peer review of "Thaw-driven mass wasting couples slopes with downstream systems and effects propagate through Arctic drainage networks."

_The Cryosphere, 2020_

## Referee Comment (RC1) · Julian Murton (Referee) · 8 Nov 2020

The aim of the manuscript is to elucidate the [geomorphic, hydrologic and, to a lesser degree, sedimentary] processes and feedbacks that drive the [decadal] evolution of thaw-related mass movements and hillslope-channel coupling in ice-rich permafrost terrain of northwest Canada (lines, L108-111). The objectives (L121-123) are: " to better understand the (A) processes that drive the intensification of thaw-driven mass wasting and slope to stream coupling, (B) the distribution of catchment effects, and (C) their propagation across watershed scales,". The aim and objectives are important directly to the fields of permafrost geomorphology and hydrology, and indirectly to the fields of biogeochemistry, terrestrial and aquatic ecology, as well as to landscape management and ecosystem services. Personally, I would rephrase the objectives as aims (because the objectives given are really general statements of intent or goals) and identify specific objectives that signpost the ways in which the aims can be achieved and evaluated (because this is clearer in assessing how successfully aims are achieved). But to some degree this is a matter of author and journal preference.

The methods used apply high-resolution three-dimensional survey techniques (light detection and ranging, and drone-based structure-from-motion) and geographical information systems (e.g. to construct digital terrain models and determine stream ordering) to drainage basins whose area varies by orders magnitude in a study region of 1 million km$^2$ in NW Canada. This allows the authors to address terrain characteristics and functional geomorphic-hydrological relationships at localized to regional scales. The methods are appropriate to the aims and objectives, and are presented clearly, systematically and rigorously as far as I can tell, though I am not an expert in GIS analysis, and so I cannot comment usefully on pages 1-8 of the supplementary information.

The results are largely new, clearly structured and presented well. The data represent a major contribution to terrain analysis on ice-rich permafrost, and the authors should be congratulated for bringing together this large and complex dataset. In particular, the focus on location of mass movements within catchments of different area, achieved through simple application of Strahler stream ordering, nicely identifies the first and second-order basins as particular centres of landscape change, and takes up functional and historical geomorphologists' consideration of scale and morphometric issues developed mainly since the 1950s in other regions. The narrative is illustrated by effective figures and tables, though some minor points need clarification (see below). The three videos provide valuable supplementary information. The length of this section is fine.

The interpretation is generally excellent, leading this reader step by step through the reasoning and contextualisation within the wider literature. The latter was particularly strong, as there has been substantial previous research on thermokarst terrain and processes in northwest Canada. The length of the interpretation could perhaps be shortened by 10-20% to avoid repetition and bring out the key messages more clearly. Likewise, the conclusions, in my view, could you shortened to a number of key points, though again I appreciate that this is a matter of preference.

Overall, I think that this manuscript makes a substantial advance in our knowledge and understanding of the impacts of thaw-related mass movements on hillslope-stream coupling ice-rich permafrost catchments in northwest Canada. The approach used could be more widely applied in other regions of ice-rich permafrost (e.g. northern Alaska, NW Siberia and NE Siberia). I recommend publication subject to mostly minor revisions concerning points of clarification and typos, as listed below. Only two points of moderate significance are raised for consideration:

**Moderate points**

**Slope thermokarst (**lines, L135-137, 195-197): Active-layer deepening and surface subsidence beneath a hillslope could reasonably be included in 'slope thermokarst', so I think this study is focussing on the most visible type of slope thermokarst, i.e. mass movement types. Perhaps this distinction can be made. 'Thaw-driven mass wasting' (L657) is a more accurate description of the focus of this manuscript than is 'slope thermokarst', in my view.

**Stabilization:** a couple of sentences might be added to comment on the contrast between the recent decadal intensification of thaw-related mass movement and the stabilization of presumably the same terrain after the early Holocene climate warming. A reader might infer from the present argument that the recent trends are here to stay, which may be true for decadal and centennial timescales, but I wonder if the early Holocene landscape suffered even more change over even longer periods (millennial), and then stabilized, preserving abundant buried ice. The

authors insights into thaw and terrain change may help elucidate negative feedbacks the thermokarst system, as Lawson, Shur and others have done successfully in terms of thermokarst around ice wedges etc.

**Minor points**

L59: Prince of Wales Strait: mark on Fig. 1

L62: North America's largest delta may be the Mississippi, a few thousand km$^2$ larger than the Mackenzie. Please check.

L77: insert 'ice-rich' into this topic sentence, because thermokarst activity will not really affect permafrost with little or no ground ice, e.g. '…evolution of circumpolar ice-rich landscapes…'

L85: specify the nature of 'Arctic change' in the topic sentence as this encompasses many things, e.g. 'of environmental change in Arctic terrestrial and aquatic systems'.

L93: 'have' [subject is plural]

L106: do you mean 'thickness' (an interval) rather than depth (a single point), i.e. permafrost thickness?

L108-113: please shorten and simplify this long, complex sentence. It's a bit difficult to follow.

L121-124: this key sentence identifies the **aims** of the study. I think it would be clearer to simplify and rephrase along the lines 'The aims of the present study are (1)…' rather than squash them into a long introductory clause. The geographical region is of secondary importance relative to the more generic aims. Also, please specify the type of processes in (A), e.g. geomorphic, thermal …, and the nature of the distribution in (B), e.g. spatial and/or temporal.

L130: append 'climate' to 'cooling Holocene'.

L141: Tuktoyaktuk Coastlands [with an 's']

L144: clarify what is meant by 'fluvial patterns', e.g. river channel morphology, bedform architecture, sediment transport…?

L145: indicate Mackenzie Delta on Fig. 1

L151: indicate Amundsen Gulf on Fig. 1

L178: replace 'middle' with 'medium'

L181-184: indicate approximate depth of mean annual ground temperatures as much of the deeper layers of permafrost on Banks and Victoria islands etc. will be much warmer than -10oC.

L192: insert 'other' before 'glacigenic materials' as tills are glacigenic.

L200: **datasets**: it's essential to identify all of the datasets used in the study rather than the non-specific word 'include'.

L210: clarify what is meant by 'a continuum of slump features'. Continuum in what sense: activity, size, aspect…?

L228: **subsidence**: did some of the volumetric change on the slopes resulted from permafrost thaw and thermokarst subsidence beneath slump-floor sediments (cf. Burn 2000, CJES 37:967–981) or can this process be discounted?

L238: **volumetric erosion**: careful, the study is not directly measuring erosion but inferring erosion based on measurement of volume change. So an explicit parameter such as 'volume change' or 'disturbance volume' (L344) is more appropriate. There may be a better term, as I'm not familiar with GIS methodology.

L240: **active or recently-active scar and debris tongues**: on what criteria were these identified as such? e.g. lack of living vegetation or some indirect evidence of vegetation? L493 mentions bare or sparsely vegetated landforms. I find it difficult to know from many GIS studies what actually is being observed directly and what is being inferred.

L245, and supplementary L277: '**including**': it is clearer to identify all of the criteria for designating a slump as '2'. Were there any other criteria besides the two mentioned, e.g. turbidity in rivers, as per caption of Figure S3ii?

L275: insert 'and' after '2010'.

L299: "All mapping was reviewed for accuracy and consistency." Please explain how or cite a reference that does.

L319: data are treated as plural in this sentence; previously (e.g. L166, 209) they are treated as singular. Please ensure consistency.

L331: routing … 'was'

L365: **intensifying** slope thermokarst: this implies that the rate of growth or the increase in number of slumps or both factors is accelerating in all cases. Is this correct for all slumps or just for some, e.g. CB?

L366: '**eroded volume**': again, can you be sure that thermal erosion accounted for all of the missing volume, or might thermokarst subsidence have contributed to the missing volume?

L383: delete 'retreat' because the photograph shows the headwall but not its retreat.

L384: Ditto 'erosion'; the photograph showed 'eroded glaciofluvial deposits', not their erosion. Also, add 's' to 'deposit'.

L387: again, the photograph in panel d does not show 'initial stages of incision', but an incised channel. Panels c and e may show evidence of side valley erosion, but they don't show any erosion itself.

L390: please indicate (e.g. with an arrow) the snow patch, as it's not obvious to me at least.

L383 & 390: please clarify the caption 'Elevation normalized debris tongue profiles…'. The y axis of the plot shows thickness, so I think this should be added to the caption, e.g. 'elevation normalized profiles of debris-tongue thickness'. Also, there are in total three white dashed lines on panels b, f and g, but four lines depicting profiles on the plot in panel h.

L394: if this refers to slump area as opposed to e.g. headwall height, then it is clearer to rewrite, e.g. '…the area of FM2 was an order of magnitude greater than …'

L397-8: 'Increasing thaw-driven sediment flows…': please clarify if this refers to their number, magnitude, rate or …

L401-402: 'pinning of the stream channel to the valley wall (Fig. 2c)': please indicate this (e.g. with an arrow) on Fig. 2c, as it's not very clear to me where the stream channel is.

L402: better to replace 'massive deposits' with 'thick deposits', as the former, in the context of sedimentary deposits, suggests that they lack sedimentary structures, which may or not be the case, as they are not described.

L406: 'abrupt transition from small valley-side thaw slumps into larger, more dynamic features': I'm not sure that the data on area of slump CB support this, as within 7-9 years of slump initiation CB was 25,900 m$^2$ (by 2011), i.e. growing at a few thousand m$^2$ per year, whereas 4 years later it was 33,370 m$^2$, which suggests a broadly similar rate of expansion. What does look to have been abrupt, is the sudden evacuation of slump-floor deposits since 2017.

L434: delete comma after 'geomorphology'

L493, 525-6 and Fig. 6: **large translational slides**: what criteria are used to identify these landforms and to distinguish them from thaw slumps? Are they different from active-layer detachments? How do you identify bedrock control?

L495: Fig. 5a is first mentioned after Fig. 6 (L493). Please correct numbering.

L498: depth of maximum thaw: how is this value determined? Do you mean the maximum concavity depth in L506?

L511: 'from 2002 to 2018'

L515-16, 739-41: "Normalizing by catchment area and differencing with the preceding time interval, the thaw slump component of surface lowering amounts to 0.1 mm yr-1 for 1986-2002 and 0.8 mm yr-1 for 2002-2018." This seems to me to be a rather strange and spurious parameter to calculate because surface lowering in thaw-slump terrain is not uniformly distributed, but focussed in discrete locations. An alternative, perhaps more meaningful parameter to

calculate would be volume lost per unit area (cf. sediment yield), because this does not imply that the lost volume is uniformly distributed across space.

L529: were [km are plural]

L546: box and whisker plots: please state what each part shows, e.g. horizontal line denotes median, … dots indicate outliers …

L547: narrative specifies 'concavity thicknesses' whereas Y axis on panel f is … 'depth'. Please ensure consistency.

L555: the proportional circles are grey rather than black.

L559-560: 'Willow Lake (outlined in Orange)': where is this on panel c? Lower case for orange or simply add a label 'Willow Lake'. Please renumber panels to avoid three panels all labelled b, and three labelled c. 'The abandoned channel is shown in dark blue': in panel c the lakes look to be coloured dark blue in Fig. 6. Or are you referring to the unlabelled panel? This is difficult to follow the caption without sequential labels on all panels and text placed accordingly.

L586-591: This summary of literature is more appropriate for a discussion than a results section.

L589: indicates (with 's'; compilation is singular)

L602 & 604: both 2017 and 2018 are indicated in caption but only 2017 is shown on panels a to c.

L606: streams and rivers: what is the difference? Insert 'of' before 'the Peel…'

L661: suggest [plural subject]

L672: '**rapid aggradation of channel beds**': perhaps 'rapid aggradation of valley fills or sediment bodies' is more appropriate. Deposition of the valley fill in Fig 2 looks to have been mainly by debris-flow processes rather than channel processes (cf. L716-20). The channel shown in Fig. 2d has incised its bed.

L693: is complex [subject is singular]

L752: is magnified [subject ('the significance') is singular]

L775: what is a 'discordant volume'?

L823: 'persistent perturbation': please specify the timescale of persistent or omit. Over decadal and possibly centennial scales, the perturbation may well be persistent. But geologically (multi-millennial and longer scales), the perturbation is certainly major but transitory, as the conceptual framework proposed by Ryder, Church, Ballantyne and others infers a pulse of sediment movement that declines over time.

L839: again, please clarify what timescales are referred to as 'long-term'. L844 identifies centennial timescales.

SuppL381: correct to 'cloud-free'

---

## Referee Comment (RC2) · Ingmar Nitze (Referee) · 14 Dec 2020

General Comments The manuscript "Permafrost thaw couples slopes with downstream systems and effects propagate through Arctic drainage networks." provides a comprehensive overview of the extent and effects of mass wasting processing in NW Canada on its associated drainage networks across different scales. It analyzes different scales from local watersheds to the entire study area of ca. 1 Mkm$^2$. The authors used numerous methodologies and data sources were applied for each specific scale and target objective. The authors did a great job. This manuscript is of high quality and very comprehensive with a lot of detail and only needs minor corrections. Here are some general

remarks. Detailed comments are stated below. The analysis of many different aspects, with a plethora of datasets in different scales, makes it sometimes hard to follow. I think it is generally very hard to find the balance between details and the overall story. Perhaps minor improvements, such as adding a table of datasets (see detailed comments) will help the readers to understand the scale, objective and significance of the specific analyses. The quality of figures ranges from very good to "room for improvement". Please check detailed comments. Geospatial datasets (Shapefiles or KML) of e.g. the slumps, and perhaps other features as well, would be a helpful addition for readers to easily find the locations and cross-check with other data sources. Overall this manuscript will be a great contribution to the permafrost science community.

Specific Comments Title: The title is rather complicated, particularly reading it for the first time 175: It would be good to somehow provide the exact number, especially since you do that in the abstract. 201 ff: You used several different datasets, but it is rather hard to follow this part with text only. I suggest to add a table with basic methodologies and datasets and its related objectives. This will help to keep better track of used methods and spatial scales. 238: Could you provide a little bit more information who exactly digitized the slumps (how many different people, people with field experience, etc.). I personally find it very challenging to consistently digitize thaw slumps, and even more so with several people. 258: I think it would make a great supplementary figure to show some examples (e.g. cross sections) of the reconstruction for selected sites. It looks like you provided this in Figure 3e, but did not reference it in the text. 267: Which Sentinel exactly. I suppose you mean Sentinel-2. 366: "decade": Do you have exact initiation ages or is it rather an estimate? If the former I suggest using a more precise values (year) otherwise it's also fine to leave decade. 369: see decades 369: I suggest writing "2" as a word, as a few words later. 376 Table 1: Please use either negative values without direction (W) or positive values with direction for longitudes. E.g. -135.7555° OR 135.7555°W 381: Figure 2: It would be nice to somehow make a more efficient use of this figure in the next version in case this will be a full page figure, as there is a large blank space on the right. Of course I understand that this version of the manuscript is

[Figure]

Interactive
comment

still a preprint. Figure 2h: This plot is somehow hard to understand at the first glace. X-Label: is not initially clear, which distance you mean. I suggest extending it to "Distance from <location>" (fill with your reference location). The same (to a lesser extent) applies to the Y-Label. I suggest using "Thickness of sediment accumulation" or so. 425 Figure 3: a-c: I suggest using a more appropriate colorbar and visual scaling with a distinct break at 0 (zero). E.g. greenish/blueish colors for accumulation and orange/reddish colors for erosion. (e.g. https://colorbrewer2.org/#type=diverging&scheme=RdBu&n=9 or something similar) a-d: What is the source and timing of the hillshade? e: (very gentle) gridlines may help to better read the proportions of the plot. However, I am not sure if this add too much information to this plot. The intersection of "(e)" and the lines may need some improvement. 434: Please specify what exactly you mean with thaw-slump indices. Volume, area, . . .? (I found them in 443). I suggest using them here. 434 ff: You are providing slump related statistics, but it is unclear which total area you analyzed with this dataset or how many features/slumps you detected. At least I cannot find them here in this paragraph. 465: I suggest using "Scatter plots" instead of only "Scatter" (if you mean scatter plots)

543 Figure5: a Inset: The numbers are hard to read especially in the dark grey part. The order in 1986 is reversed (2 bottom) compared to the other years. Technically you could include the same information into the large bars, though the focus shifts to absolute numbers rather than relative to 100%. b-c/d-e: As the information from b & c as well as d & e are highly correlated, using only area or volume might be appropriate. However, this is a "soft" recommendation, but might be ok if you leave it. Perhaps shifting f to position d makes sense to have area and volume in one column.

551 Figure 6: Please check the numbering (a-c) of the insets. The main map does not have a letter. There are duplicate b and c. Main map: Just out of curiosity, which projection is it? The maps seems to be slightly rotated (clock-wise). The grid shows the rotation but the north arrow does not. Please adapt the north arrow, as I suppose the rotation was made to fit the watershed into the figure. I like the accumulated scar
area visualization.

567: Sentinel-2? 596 Figure 7: The data itself are very interesting but the visualization should be improved. I suggest using colors instead of black and white only. Furthermore please make sure that data is not occluded, particularly in a, b and d. Using colors and semi-transparent markers should help. Is a/b already semi-transparent and the grey part the intersection area? If yes, using colors will help to better see that this might be the intersecting area, as this color is not visible in the legend. Perhaps, you could remove the fill color for the bars at all and use only edge colors. b/c: These plots look good, but it's quite challenging to understand what they mean. Particularly c it is not clear to me what the Cumulative disturbance in relation to the catchment area means. I see that there are changes over time (Peel), but the specific data behind it are puzzling to me.

641 Figure 8: Awesome Figure What does NHN mean in the legend?

859: I think it will help to have a list of publically available datasets in this section have a direct and comprehensive overview of these datasets instead of crawling through the text and references. Access to your datasets, e.g. delineated thaw slumps or aggregated spatial statistics, will be of great benefit to other researchers, particularly for large scale remote sensing and model applications.

Supplement Figure S1: I suggest using colors for nicer visualization. Please add (a,b,c) to each subplot. A horizontal alignment of plots would be nice, even if the plot size needs to be slightly reduced. Figure S3 iii: Here it would be great if you'd add an arrow/marker to the slide. With a lot of experience it's possible to find it, but without it can be hard. Please mention (and visually indicate) the dam/blockage. 384: bottom of Table S3: Perhaps it is rather nitpicky, but using ISO format of dates would be nicer (e.g. 1986_07_07 –> 1986-07-07) 444ff: Perhaps you should cite https://jstnbraaten.shinyapps.io/snazzy-ee-ts-gif/ as well, which is the second step to create these animations.

Technical Comments Supplement 381 Table S3 caption: typo in: "... could free Landsat ..."

Please also note the supplement to this comment:
https://tc.copernicus.org/preprints/tc-2020-218/tc-2020-218-RC2-supplement.pdf

---

## Author Comment (AC1) · 15 Feb 2021

AUTHOR REPLY We are grateful for Dr. Murton's detailed and thoughtful review and provide replies to his comments in the following document. We feel that Dr. Murton's comments have improved the clarity of the manuscript and we were able to implement his recommended changes through minor editorial modifications to the text and some figures. Dr. Murton's contribution to improving our manuscript is recognized in the Acknowledgements section.

R1. Dr. Julian Murton The aim of the manuscript is to elucidate the [geomorphic, hydrologic and, to a lesser degree, sedimentary] processes and feedbacks that drive the

[decadal] evolution of thaw-related mass movements and hillslope-channel coupling in ice-rich permafrost terrain of northwest Canada (lines, L108-111). The objectives (L121-123) are: " to better understand the (A) processes that drive the intensification of thaw-driven mass wasting and slope to stream coupling, (B) the distribution of catchment effects, and (C) their propagation across watershed scales,". The aim and objectives are important directly to the fields of permafrost geomorphology and hydrology, and indirectly to the fields of biogeochemistry, terrestrial and aquatic ecology, as well as to landscape management and ecosystem services. Personally, I would rephrase the objectives as aims (because the objectives given are really general statements of intent or goals) and identify specific objectives that signpost the ways in which the aims can be achieved and evaluated (because this is clearer in assessing how successfully aims are achieved). But to some degree this is a matter of author and journal preference. The methods used apply high-resolution three-dimensional survey techniques (light detection and ranging, and drone-based structure-from-motion) and geographical information systems (e.g. to construct digital terrain models and determine stream ordering) to drainage basins whose area varies by orders magnitude in a study region of 1 million km2 in NW Canada. This allows the authors to address terrain characteristics and functional geomorphic-hydrological relationships at localized to regional scales. The methods are appropriate to the aims and objectives, and are presented clearly, systematically and rigorously as far as I can tell, though I am not an expert in GIS analysis, and so I cannot comment usefully on pages 1-8 of the supplementary information. The results are largely new, clearly structured and presented well. The data represent a major contribution to terrain analysis on ice-rich permafrost, and the authors should be congratulated for bringing together this large and complex dataset. In particular, the focus on location of mass movements within catchments of different area, achieved through simple application of Strahler stream ordering, nicely identifies the first and second-order basins as particular centres of landscape change, and takes up functional and historical geomorphologists' consideration of scale and morphometric issues developed mainly since the 1950s in other regions. The narrative is illustrated

Interactive
comment

by effective figures and tables, though some minor points need clarification (see below). The three videos provide valuable supplementary information. The length of this section is fine. The interpretation is generally excellent, leading this reader step by step through the reasoning and contextualisation within the wider literature. The latter was particularly strong, as there has been substantial previous research on thermokarst terrain and processes in northwest Canada. The length of the interpretation could perhaps be shortened by 10-20% to avoid repetition and bring out the key messages more clearly. Likewise, the conclusions, in my view, could you shortened to a number of key points, though again I appreciate that this is a matter of preference. Overall, I think that this manuscript makes a substantial advance in our knowledge and understanding of the impacts of thaw-related mass movements on hillslope-stream coupling ice-rich permafrost catchments in northwest Canada. The approach used could be more widely applied in other regions of ice-rich permafrost (e.g. northern Alaska, NW Siberia and NE Siberia). I recommend publication subject to mostly minor revisions concerning points of clarification and typos, as listed below. Only two points of moderate significance are raised for consideration:

REPLY to comments by Dr. Julian Murton (R1) Following the recommendation of Dr. Murton we now refer to our statements of project intent as "Aims" or "Goals". We have also endeavored to cut the length of the discussion and conclusions sections to avoid redundancy by about 10%. Minor elaborations required to address a few reviewer comments made a greater reduction of manuscript length difficult.

Moderate points Slope thermokarst (lines, L135-137, 195-197): Active-layer deepening and surface subsidence beneath a hillslope could reasonably be included in 'slope thermokarst', so I think this study is focussing on the most visible type of slope thermokarst, i.e. mass movement types. Perhaps this distinction can be made. 'Thaw-driven mass wasting' (L657) is a more accurate description of the focus of this manuscript than is 'slope thermokarst', in my view.

REPLY. We agree that slope thermokarst could indicate processes that extend beyond

mass wasting. In the introduction, we clarify that the thaw-driven mass wasting process including retrogressive thaw slumping, shallow and deep translational failures comprise the most dynamic forms of slope thermokarst and that these processes are the focus of our study (L104). In some cases, we still use the term "slope thermokarst", for instance where we make general statements such as "climate-driven intensification of slope thermokarst". We have made numerous adjustments throughout the manuscript and refer explicitly to thaw slumps when addressing this specific process, and to thaw-driven mass wasting, permafrost landslides, or thaw-driven landslides when referring to a wider suite of processes that are included in broad-scale fluvial network analysis (L133).

Stabilization: a couple of sentences might be added to comment on the contrast between the recent decadal intensification of thaw-related mass movement and the stabilization of presumably the same terrain after the early Holocene climate warming. A reader might infer from the present argument that the recent trends are here to stay, which may be true for decadal and centennial timescales, but I wonder if the early Holocene landscape suffered even more change over even longer periods (millennial), and then stabilized, preserving abundant buried ice. The authors insights into thaw and terrain change may help elucidate negative feedbacks the thermokarst system, as Lawson, Shur and others have done successfully in terms of thermokarst around ice wedges etc.

REPLY. We are not overly comfortable speculating on the negative feedbacks in great detail as it is beyond the scope of this study. Here we focus on contemporary processes and on providing a geological explanation for rates and patterns of slope thermokarst acceleration. The evolution of negative feedbacks is likely to be of relevance over time scales greater than a century. However, we acknowledge Dr. Murton's point and feel that it is a topic for further consideration in research examining conditions in the late Holocene through paleoenvironmental methods. We have added a sentence in the discussion addressing Dr. Murton's point on L783-L786 "Several feedbacks could

counteract the present intensification of thaw-driven mass wasting and increasing sed-
imentary and geochemical fluxes including climate cooling, exhaustion of sediment
supply and progressive loss of ground ice from the most sensitive slopes, and gradual
thaw-driven decreases in slope gradients, however, these factors are likely to be most
relevant at centennial time-scales or greater."

Minor points L59: Prince of Wales Strait: mark on Fig. 1 REPLY. Changed as sug-
gested

L62: North America's largest delta may be the Mississippi, a few thousand km2 larger
than the Mackenzie. Please check. REPLY. Minor editorial adjustment made indicating
"…North America's largest Arctic delta and the Beaufort Sea."

L77: insert 'ice-rich' into this topic sentence, because thermokarst activity will not really
affect permafrost with little or no ground ice, e.g. '…evolution of circumpolar ice-rich
landscapes…' REPLY. Changed as suggested.

L85: specify the nature of 'Arctic change' in the topic sentence as this encompasses
many things, e.g. 'of environmental change in Arctic terrestrial and aquatic systems'.
REPLY. Changed as suggested.

L93: 'have' [subject is plural] REPLY. Changed as suggested.

L106: do you mean 'thickness' (an interval) rather than depth (a single point), i.e.
permafrost thickness? REPLY. Changed as suggested.

L108-113: please shorten and simplify this long, complex sentence. It's a bit difficult to
follow. REPLY. Changed through editorial modification.

L121-124: this key sentence identifies the aims of the study. I think it would be clearer
to simplify and rephrase along the lines 'The aims of the present study are (1)…'
rather than squash them into a long introductory clause. The geographical region is of
secondary importance relative to the more generic aims. Also, please specify the type
of processes in (A), e.g. geomorphic, thermal …, and the nature of the distribution in

(B), e.g. spatial and/or temporal. REPLY. Changed through minor editorial modification.

L130: append 'climate' to 'cooling Holocene'. REPLY. Changed as suggested.

L141: Tuktoyaktuk Coastlands [with an 's'] REPLY. Changed as suggested.

L144: clarify what is meant by 'fluvial patterns', e.g. river channel morphology, bedform architecture, sediment transport...? REPLY. Changed through minor editorial modification.

L145: indicate Mackenzie Delta on Fig. 1 REPLY. Changed as suggested.

L151: indicate Amundsen Gulf on Fig. 1 REPLY. Changed as suggested.

L178: replace 'middle' with 'medium' REPLY. Changed as suggested.

L181-184: indicate approximate depth of mean annual ground temperatures as much of the deeper layers of permafrost on Banks and Victoria islands etc. will be much warmer than -10oC. REPLY. Depth reference is now indicated as the mean annual temperature at the top of permafrost (TTOP).

L192: insert 'other' before 'glacigenic materials' as tills are glacigenic. REPLY. Changed as suggested.

L200: datasets: it's essential to identify all of the datasets used in the study rather than the non-specific word 'include'. REPLY. A new supplementary table has been added (Table S1) to make explicit the datasets used in this study and the research questions that they address. "To examine processes driving the intensification of thaw-driven mass wasting and the patterns of effects across hydrological networks we applied multiple methods involving field study and mapping at slope, catchment, and watershed scales described in the following sections and summarized in Table S1."

L210: clarify what is meant by 'a continuum of slump features'. Continuum in what sense: activity, size, aspect...? REPLY. Minor editorial adjustment made to clarify that the slumps represent a size continuum.

L228: subsidence: did some of the volumetric change on the slopes resulted from permafrost thaw and thermokarst subsidence beneath slump-floor sediments (cf. Burn 2000, CJES 37:967–981) or can this process be discounted? REPLY. We make a minor editorial adjustment to clarify that volume changes within the scar zone include both thaw-subsidence and slope erosion.

L238: volumetric erosion: careful, the study is not directly measuring erosion but inferring erosion based on measurement of volume change. So an explicit parameter such as 'volume change' or 'disturbance volume' (L344) is more appropriate. There may be a better term, as I'm not familiar with GIS methodology. REPLY. We replaced "erosional" with "disturbance".

L240: active or recently-active scar and debris tongues: on what criteria were these identified as such? e.g. lack of living vegetation or some indirect evidence of vegetation? L493 mentions bare or sparsely vegetated landforms. I find it difficult to know from many GIS studies what actually is being observed directly and what is being inferred. REPLY. We clarify in the text that active or recently-active scar and debris tongues were interpreted by a distinct scarp and bare or sparsely vegetated scar area determined with the support of high-resolution orthomosaic imagery.

L245, and supplementary L277: 'including': it is clearer to identify all of the criteria for designating a slump as '2'. Were there any other criteria besides the two mentioned, e.g. turbidity in rivers, as per caption of Figure S3ii? REPLY. The text is clarified to indicate that evidence of downstream deposition is expressed as a debris tongue deposit in a valley bottom, or a sediment lobe protruding into an adjacent lake or coastline.

L275: insert 'and' after '2010'. REPLY. Changed as suggested.

L299: "All mapping was reviewed for accuracy and consistency." Please explain how or cite a reference that does. REPLY. Reference is provided to Kokoszka and Kokelj, 2020, which provides a detailed description of methodology and QA/QC procedures.

L319: data are treated as plural in this sentence; previously (e.g. L166, 209) they are treated as singular. Please ensure consistency. REPLY. Minor adjustments on L166 and 209 ensure that data are treated as plural.

L331: routing . . . 'was' REPLY. Changed as suggested.

L365: intensifying slope thermokarst: this implies that the rate of growth or the increase in number of slumps or both factors is accelerating in all cases. Is this correct for all slumps or just for some, e.g. CB? REPLY. Minor editorial modification was implemented to clarify the sentence.

L366: 'eroded volume': again, can you be sure that thermal erosion accounted for all of the missing volume, or might thermokarst subsidence have contributed to the missing volume? REPLY. We clarify that volume loss associated with thaw slump development is a function of both sediment erosion and subsidence due to ground ice thaw.

L383: delete 'retreat' because the photograph shows the headwall but not its retreat. REPLY. Changed as suggested. Please note that based on comments from both Reviewers we have rearranged Figure 2 and the caption to increase its clarity. We have added elevation normalized slope profiles to the debris tongue profiles to better illustrate the connection between thaw-driven erosion and deposition. We also have created a supplementary figure (S1) to show terrain models of the slumps and locations of these transects.

L384: Ditto 'erosion'; the photograph showed 'eroded glaciofluvial deposits', not their erosion. Also, add 's' to 'deposit'. REPLY. Changed as suggested.

L387: again, the photograph in panel d does not show 'initial stages of incision', but an incised channel. Panels c and e may show evidence of side valley erosion, but they don't show any erosion itself. REPLY. Changed as suggested.

L390: please indicate (e.g. with an arrow) the snow patch, as it's not obvious to me at least. REPLY. Caption was adjusted to increase clarity and reference to the snowpatch

was removed.

L383 & 390: please clarify the caption 'Elevation normalized debris tongue profiles. . .'. The y axis of the plot shows thickness, so I think this should be added to the caption, e.g. 'elevation normalized profiles of debris-tongue thickness'. Also, there are in total three white dashed lines on panels b, f and g, but four lines depicting profiles on the plot in panel h. REPLY. Figure 2 has been adjusted to increase its clarity and white dashed lines have been removed.

L394: if this refers to slump area as opposed to e.g. headwall height, then it is clearer to rewrite, e.g. '. . .the area of FM2 was an order of magnitude greater than . . .' REPLY. Changed as suggested.

L397-8: 'Increasing thaw-driven sediment flows. . .': please clarify if this refers to their number, magnitude, rate or . . . REPLY. Minor editorial adjustment was implemented to clarify the text.

L401-402: 'pinning of the stream channel to the valley wall (Fig. 2c)': please indicate this (e.g. with an arrow) on Fig. 2c, as it's not very clear to me where the stream channel is. REPLY. Figure 2 has been adjusted and photograph was removed.

L402: better to replace 'massive deposits' with 'thick deposits', as the former, in the context of sedimentary deposits, suggests that they lack sedimentary structures, which may or not be the case, as they are not described. REPLY. Changed as suggested.

L406: 'abrupt transition from small valley-side thaw slumps into larger, more dynamic features': I'm not sure that the data on area of slump CB support this, as within 7-9 years of slump initiation CB was 25,900 m2 (by 2011), i.e. growing at a few thousand m2 per year, whereas 4 years later it was 33,370 m2, which suggests a broadly similar rate of expansion. What does look to have been abrupt, is the sudden evacuation of slump-floor deposits since 2017. REPLY. A slight adjustment in the topic sentence was implemented to better characterize the transition in thaw-driven slope evolution

observed in the study area. "The abrupt transition of shallow valley-side thaw slumps into more dynamic failures connected to downstream environments is transforming the geomorphology of ice-rich glaciated landscapes."

L434: delete comma after 'geomorphology' REPLY. Changed as suggested.

L493, 525-6 and Fig. 6: large translational slides: what criteria are used to identify these landforms and to distinguish them from thaw slumps? Are they different from active-layer detachments? How do you identify bedrock control? REPLY. The morphology of rotational or translational failures differs from thaw slumps. Field observations throughout the Willow River catchment confirmed the interpretation of remotely sensed imagery. The text has been adjusted accordingly.

L495: Fig. 5a is first mentioned after Fig. 6 (L493). Please correct numbering. REPLY. Changed as suggested.

L498: depth of maximum thaw: how is this value determined? Do you mean the maximum concavity depth in L506? REPLY. Changed to "maximum concavity depth".

L511: 'from 2002 to 2018' REPLY. Changed as suggested.

L515-16, 739-41: "Normalizing by catchment area and differencing with the preceding time interval, the thaw slump component of surface lowering amounts to 0.1 mm yr-1 for 1986-2002 and 0.8 mm yr-1 for 2002-2018." This seems to me to be a rather strange and spurious parameter to calculate because surface lowering in thaw-slump terrain is not uniformly distributed, but focussed in discrete locations. An alternative, perhaps more meaningful parameter to calculate would be volume lost per unit area (cf. sediment yield), because this does not imply that the lost volume is uniformly distributed across space. REPLY. We have reported as a surface lowering amount (mm yr-1) and also in terms of sediment yield (m3 km-2 yr-1). We initially did not report the latter (cf. sediment yield) because it too integrates estimates across the entire catchment source area and assumes that materials are getting to a catchment outlet. Regardless, we

now report both values and remind readers that a significant portion of the volumes is attributed to ground ice (50 to 80%) and that a large portion of the sediments mobilized from thawing slopes are placed into transient storage in valley bottoms (see Fig. 2).

L529: were [km are plural] REPLY. Changed as suggested.

L546: box and whisker plots: please state what each part shows, e.g. horizontal line denotes median, ... dots indicate outliers ... REPLY. Minor editorial adjustment implemented to clarify text.

L547: narrative specifies 'concavity thicknesses' whereas Y axis on panel f is ... 'depth'. Please ensure consistency. REPLY. Changed as suggested.

L555: the proportional circles are grey rather than black. REPLY. Changed as suggested

L559-560: 'Willow Lake (outlined in Orange)': where is this on panel c? Lower case for orange or simply add a label 'Willow Lake'. Please renumber panels to avoid three panels all labelled b, and three labelled c. 'The abandoned channel is shown in dark blue': in panel c the lakes look to be coloured dark blue in Fig. 6. Or are you referring to the unlabelled panel? This is difficult to follow the caption without sequential labels on all panels and text placed accordingly. REPLY. We have adjusted the caption and made minor edits to the figure. All elements of the main map are indicated in the first part of the caption, including reference to the inset boxes. The panels are labeled sequentially and explained in the caption.

L586-591: This summary of literature is more appropriate for a discussion than a results section. REPLY. References are removed from the text.

L589: indicates (with 's'; compilation is singular) REPLY. Changed as suggested.

L602 & 604: both 2017 and 2018 are indicated in caption but only 2017 is shown on panels a to c. REPLY. Editorial clarification implemented. The Sentinel-2 data are for 2016-2017 and text and figures have been checked for consistency.

[Figure]

606: streams and rivers: what is the difference? Insert 'of' before 'the Peel. . .' REPLY. Changed to "streams".

L661: suggest [plural subject] REPLY. Changed as suggested.

L672: 'rapid aggradation of channel beds': perhaps 'rapid aggradation of valley fills or sediment bodies' is more appropriate. Deposition of the valley fill in Fig 2 looks to have been mainly by debris-flow processes rather than channel processes (cf. L716-20). The channel shown in Fig. 2d has incised its bed. REPLY. Changed as suggested.

L693: is complex [subject is singular] REPLY. Change as suggested.

L752: is magnified [subject ('the significance') is singular] REPLY. Changed as suggested.

L775: what is a 'discordant volume'? REPLY. Minor editorial modification implemented to clarify text.

L823: 'persistent perturbation': please specify the timescale of persistent or omit. Over decadal and possibly centennial scales, the perturbation may well be persistent. But geologically (multi-millennial and longer scales), the perturbation is certainly major but transitory, as the conceptual framework proposed by Ryder, Church, Ballantyne and others infers a pulse of sediment movement that declines over time. REPLY. Minor editorial modification implemented to provide a timescale (decadal to millennial) over which perturbations are likely to persist.

L839: again, please clarify what timescales are referred to as 'long-term'. L844 identifies centennial timescales. REPLY. Minor editorial modification implemented to provide a timescale (centennial to millennial) over which perturbations are likely to persist.

SuppL381: correct to 'cloud-free' REPLY. Change as suggested.

---

## Author Comment (AC2) · 15 Feb 2021

Reply to comments by Dr. Ingmar Nitze (R2)

We appreciate the thoughtful input provided by Dr. Nitze and have undertaken several minor modifications to address his suggestions. In particular, constructive critique of several figures has resulted in improvements that have increased their clarity and impact. Dr. Nitze's contribution to improving our manuscript is recognized in the Acknowledgements section. Detailed replies are provided to specific comments below.

General Comments The manuscript "Permafrost thaw couples slopes with downstream

[Figure]

systems and effects propagate through Arctic drainage networks." provides a comprehensive overview of the extent and effects of mass wasting processing in NW Canada on its associated drainage networks across different scales. It analyzes different scales from local watersheds to the entire study area of ca. 1 Mkm2 . The authors used numerous methodologies and data sources were applied for each specific scale and target objective. The authors did a great job. This manuscript is of high quality and very comprehensive with a lot of detail and only needs minor corrections.

Here are some general remarks. Detailed comments are stated below. The analysis of many different aspects, with a plethora of datasets in different scales, makes it sometimes hard to follow. I think it is generally very hard to find the balance between details and the overall story. Perhaps minor improvements, such as adding a table of datasets (see detailed comments) will help the readers to understand the scale, objective and significance of the specific analyses. The quality of figures ranges from very good to "room for improvement". Please check detailed comments. Geospatial datasets (Shapefiles or KML) of e.g. the slumps, and perhaps other features as well, would be a helpful addition for readers to easily find the locations and cross-check with other data sources. Overall this manuscript will be a great contribution to the permafrost science community.

REPLY. We acknowledge that the paper covers a wide range of scales as our goals are to link processes on slopes, evolving connectivity with downstream environments and propagation of effects across Arctic drainage networks. To help provide readers with great clarity on what datasets were utilized in the paper and what research questions they address we have constructed Table S1. The table indicates main research objectives and identifies and describes the datasets used to address these questions and their source. We have also made several minor adjustments to figures or captions where possible to improve their quality and clarity. Finally, shapefiles will be published with the associated Open Reports that are referenced in Table S1, which have been reviewed and will be released concurrently with paper publication through the Northwest

Territories Geological Survey Open Report system.

Specific Comments Title: The title is rather complicated, particularly reading it for the first time

REPLY. We have made a slight modification to the title to improve its clarity.

175: It would be good to somehow provide the exact number, especially since you do that in the abstract.

REPLY. Changed as suggested.

201 ff: You used several different datasets, but it is rather hard to follow this part with text only. I suggest to add a table with basic methodologies and datasets and its related objectives. This will help to keep better track of used methods and spatial scales.

REPLY. Table S1 was developed to indicate the datasets utilized, the research question they pertain to, the base data from which interpretations were made, and reference to data sources.

238: Could you provide a little bit more information who exactly digitized the slumps (how many different people, people with field experience, etc.). I personally find it very challenging to consistently digitize thaw slumps, and even more so with several people.

REPLY. The Authors of the paper did all of the digitizations and have field experience in the study area. It should be noted that to address various research questions in this paper, various base-data layers were used, such that delineation rules and resolution of digitized outputs will vary even if the same individual is mapping features. Methodological aspects of slump digitization is beyond scope of this paper, however, co-author Jurjen van der Sluijs is leading a critical assessment of slump delineation and classification within the context of assessing scatter within area-volume relationship functions. The majority of the datasets used in this paper are published as open reports and the Authors of those reports were responsible for digitization. Also, some examples of the base data and slump digitization are provided in new Figure S1.

258: I think it would make a great supplementary figure to show some examples (e.g. cross sections) of the reconstruction for selected sites. It looks like you provided this in Figure 3e, but did not reference it in the text.

REPLY. Developing these images are of interest but beyond the scope of this paper. The terrain reconstruction methods are explored in a robust manner through a separate methods paper. However, we have added supplementary Figure S1 which shows DTMs of thaw slumps discussed in detail in Sect. 3.1 & 3.2 and the position of topographic profiles shown in Figure 2g

267: Which Sentinel exactly. I suppose you mean Sentinel-2.

REPLY. We have clarified that Sentinel-2 imagery was used in this analysis.

366: "decade": Do you have exact initiation ages or is it rather an estimate? If the former I suggest using a more precise values (year) otherwise it's also fine to leave decade.

REPLY. The feature is present as a very small stream side slump on the 2011 LiDAR (544 m2) suggesting it had initiated only a few years prior to that. We have left the text as is.

369: see decades 369: I suggest writing "2" as a word, as a few words later.

REPLY. Changed as suggested.

376 Table 1: Please use either negative values without direction (W) or positive values with direction for longitudes. E.g. -135.7555âŮę OR 135.7555âŮęW 381:

REPLY. Changed as suggested.

Figure 2: It would be nice to somehow make a more efficient use of this figure in the next version in case this will be a full-page figure, as there is a large blank space on the right. Of course I understand that this version of the manuscript is still a preprint. Figure 2h: This plot is somehow hard to understand at the first glace. XLabel: is not

initially clear, which distance you mean. I suggest extending it to "Distance from " (fill with your reference location). The same (to a lesser extent) applies to the Y-Label. I suggest using "Thickness of sediment accumulation" or so.

REPLY. We appreciate the suggestions to improve this figure. We have made the layout more symmetrical and show pictures of slumps across the size continuum. We have also extended the topographic profiles relative to the pre-disturbed surface from the top of the slump headwall to the end of the debris tongue in order to better capture the evolution of slope to stream connectivity associated with enlargement of disturbances. Please note that new Figure S1 shows the digital terrain models, delineations and transect locations of all thaw slumps shown in Figure 2.

425 Figure 3: a-c: I suggest using a more appropriate colorbar and visual scaling with a distinct break at 0 (zero). E.g. greenish/blueish colors for accumulation and orange/reddish colors for erosion. (e.g. https://colorbrewer2.org/#type=diverging&scheme=RdBu&n=9 or something similar) a-d: What is the source and timing of the hillshade? e: (very gentle) gridlines may help to better read the proportions of the plot. However, I am not sure if this add too much information to this plot. The intersection of "(e)" and the lines may need some improvement.

REPLY. We appreciate the suggestions and have adjusted the color scheme to improve clarity of the zero breakpoints. Timing of the hill-shade is indicated in the caption and a slight tweak to the placement of annotations was implemented to decrease clutter.

434: Please specify what exactly you mean with thaw-slump indices. Volume, area, . . .? (I found them in 443). I suggest using them here.

REPLY. Minor editorial modification was made to clarify the text.

434 ff: You are providing slump related statistics, but it is unclear which total area you analyzed with this dataset or how many features/slumps you detected. At least I cannot

find them here in this paragraph.

REPLY. Method Section (2.3) describing the region from which these slumps were selected, how they were subsampled from the entire population, and the base data from which they were digitized is now referenced.

465: I suggest using "Scatter plots" instead of only "Scatter" (if you mean scatter plots)

REPLY. We clarify by indicating that we refer to "variation in the residuals, or (scatter)".

543 Figure5: an Inset: The numbers are hard to read especially in the dark grey part. The order in 1986 is reversed (2 bottom) compared to the other years. Technically you could include the same information into the large bars, though the focus shifts to absolute numbers rather than relative to 100%. b-c/d-e: As the information from b & c as well as d & e are highly correlated, using only area or volume might be appropriate. However, this is a "soft" recommendation, but might be ok if you leave it. Perhaps shifting f to position d makes sense to have area and volume in one column.

REPLY. We appreciate the advice and have removed the inset from Figure 5a and present the connectivity by shading on the large bars to portray by count. We have left the remaining figure as it is so a reader can assess both cumulative effects of increasing slump numbers as well as changes in the size distribution of the population.

551 Figure 6: Please check the numbering (a-c) of the insets. The main map does not have a letter. There are duplicate b and c. Main map: Just out of curiosity, which projection is it? The maps seems to be slightly rotated (clock-wise). The grid shows the rotation but the north arrow does not. Please adapt the north arrow, as I suppose the rotation was made to fit the watershed into the figure. I like the accumulated scar C3 area visualization.

REPLY. We have adjusted panels so that they are indicated sequentially. Caption has also been adjusted. The orientation of the North arrow has been adjusted.

567: Sentinel-2?

REPLY. Minor edit implemented to clarify the text.

596 Figure 7: The data itself are very interesting but the visualization should be improved. I suggest using colors instead of black and white only. Furthermore please make sure that data is not occluded, particularly in a, b and d. Using colors and semi-transparent markers should help. Is a/b already semi-transparent and the grey part the intersection area? If yes, using colors will help to better see that this might be the intersecting area, as this color is not visible in the legend. Perhaps, you could remove the fill color for the bars at all and use only edge colors. b/c: These plots look good, but it's quite challenging to understand what they mean. Particularly c it is not clear to me what the Cumulative disturbance in relation to the catchment area means. I see that there are changes over time (Peel), but the specific data behind it are puzzling to me.

REPLY. Adjustment of the color scheme and use of semi-transparency was implemented to improve Figure 7a, b. We have slightly adjusted the text to clarify what the data on Figure 7c portray and its significance.

641 Figure 8: Awesome Figure What does NHN mean in the legend?

REPLY. We appreciate the feedback! We have clarified in the caption that NHN refers to National Hydro Network. The dataset and acronym are also provided in the methods section and now in the new Table S1.

859: I think it will help to have a list of publically available datasets in this section have a direct and comprehensive overview of these datasets instead of crawling through the text and references. Access to your datasets, e.g. delineated thaw slumps or aggregated spatial statistics, will be of great benefit to other researchers, particularly for large scale remote sensing and model applications.

REPLY. We appreciate the comment and have addressed this in large part through providing a summary of the datasets in Table S1.

Supplement Figure S1: I suggest using colors for nicer visualization. Please add (a,b,c)

[Figure]

to each subplot. A horizontal alignment of plots would be nice, even if the plot size needs to be slightly reduced.

REPLY. Minor adjustments were made to improve the figure (now Figure S2).

Figure S3 iii: Here it would be great if you'd add an arrow/marker to the slide. With a lot of experience it's possible to find it, but without it can be hard. Please mention (and visually indicate) the dam/blockage.

REPLY. Caption adjusted to clarify location of the slide and damming of the river. Photographs have also been added so readers can visualize the magnitude of this landslide. Similar adjustments were implemented to Figure S3ii. Please not this is now Figure S4.

384: bottom of Table S3: Perhaps it is rather nitpicky, but using ISO format of dates would be nicer (e.g. 1986_07_07 –> 1986-07-07)

REPLY. Minor adjustment implemented.

444: Perhaps you should cite https://jstnbraaten.shinyapps.io/snazzy-ee-ts-gif/ as well, which is the second step to create these animations.

REPLY. The URL is now provided.

C4 Technical Comments Supplement 381 Table S3 caption: typo in: ". . . could free Landsat . . ."

REPLY. Adjusted through minor modification to the text.

―――――――――――――――――――――――

---

## Editor Decision (ED1)

[revised manuscript text omitted]

Inset text box (comment on map):

I'm asking again, because I cannot find a response to this comment: Are some of these segments missing a circle, such as this one? How can there be a red impacted segment if there is no scar area to impact it? Shouldn't accumulated scar area start at something greater than 0? Presumably the blue segments upstream are also 0s, so they could be red, or these "0" segments should be blue.

[revised manuscript text omitted]

**Supplementary Materials include:**

**Supplementary Methods 1-3** (Geoprocessing steps for hydrological network analysis of slump effects and flow accumulation analysis).

**Supplementary Figures 1-4.**

**Supplementary Tables 1-5.**

**Supplementary Videos 1-3.** (URL links provided to SV 1 and 2. Video 3 is attached).

**Supplementary Methods 1-3**

**Supplementary Method 1. Workflow for Willow River downstream accumulation of effects.**

To quantify the scar and debris tongue areas of slope thermokarst features (STKs) affecting streams in the Willow River catchment and the downstream accumulation of STK scar areas through the fluvial network as an index for sedimentary and geochemical impact (main text Sect. 2.5.), we used delineations of STK scar/debris areas (Rudy and Kokelj, 2020), hydrologic data (streams and lakes) from the 1:50,000 National Hydro Network (NHN) dataset (Natural Resources Canada,

2016), and RivEX 10.25 software (Hornby, 2017). Data were compiled in ArcMap 10.6 and projected in the Canada Lambert Conformal Conic (CLCC) projected coordinate system. The workflow is documented in the following steps:

1) Manually delineated the Willow River catchment, in ArcMap 10.6, using topographic data from the 1:50,000 Canadian Digital Elevation Model (CDED) dataset (Natural Resources Canada, 2015). In order to aid the interpretation of the watershed boundary, NHN Primary Directed Network Linear Flow (PDNLF) and Waterbody

Shapefiles (10MC002) were added to represent streams and lakes, respectively.

2) Clipped PDNLF and Waterbody feature classes (10MC002) to the Willow River catchment delineation from step 1.

3) Manually adjusted the PDNLF features to accommodate a recent channel abandonment and alteration in the routing of stream flow in the lower part of the catchment (main text Sect. 2.5.). Adjustments included:

    a) Removed PDNLF polyline (nid: 186964fe0c0c425c82150c486f7cbb71).

b) Split PDNLF polyline (nid: a44c53c852f14ff29561a048435aca29).

    c) Added PDNLF polylines and assigned nid's: a1, a2, and a3.

    d) Reversed the direction of PDNLF polylines with the following nids:

        i) 8e0462e45626485bb69e6a7932f30c32

        ii) 9cb7a1ca015543a5ab825684402188b0

 iii) 6897bbd8a58c468c8cd15caf0610f5c2

        iv) D94bcd35f20d46f39d9f920b5e4e5365

        v) 92b84230ed9a4f6e9a881a18effa8884

        vi) f53ff86131864f3ea2adcfc8dd34b780

4) Constructed a topological network and ran 'Quality Control' tools using RivEX. Adjusted PDNLF polylines as necessary to ensure network continuity (i.e. consistent from- and to-nodes).

5) NHN PDNLF features represent the primary or (main route) of a stream. However, the PDNLF features are segmented in sections of braided channels that result in pseudo-nodes. To remove pseudo-nodes in sections of braided channels, a pseudo-node free network was generated using the 'Create Network Free of Pseudo Nodes' tool in RivEX.

6) Generated points at the intersections of streams and lakes and split the streams at the intersection points using a search radius of 5m.

7) Using the pseudo-node free network from step 6, constructed a topological network, using RivEx, and ran 'Quality Control' tools (except monotonic trends). During construction of the topological network, RivEX assigned each PDNLF polyline (segment) a unique ID (RivID).

8) Added a field (RivID) to the STK scar and debris shapefiles, from Rudy and Kokelj (2020), with the field type assigned as 'short integer'.

9) Assessed STK affects to stream and lake features, using delineations of STK scar and debris areas derived from 2018 Landsat imagery. STK features were interpreted to affect streams or lakes based on direct contact with the hydrological feature or based on the direction of down-slope flow indicated by topographic data (i.e. CDED).

PDNLF segments affected by STK were recorded by assigning the value of the 'RivID' field, from the PDNLF segment, to the corresponding STK polygon. Special considerations included:

   a) Where STK(s) affected a lake, the RivID that corresponded with the hydrological/stream segment, located within the lake and at the lake outflow, was assigned to the STK polygon.

   b) In cases where STK affected multiple stream segments, the RivID from the most upstream segment, was
assigned to the STK polygon. Some exceptions included:

      i) Where STK affected both a main-stem and a tributary the RivID from the stream segment with the largest contact length, with the STK polygon, was assigned to the STK polygon.

      ii) Where STK affected multiple headwater streams, the RivID from the stream segment with the largest contact length, with the STK polygon, was assigned to the STK polygon.

10) In order to ensure the consistency of areal measurements from the research paper, the area of delineations for scar and debris areas, using 2018 Landsat, were computed using the Universal Transverse Mercator Zone 8, North American Datum 1983 (NAD 83) coordinate system.

11) Using ArcMap's 'Summary Statistics' tool, a summary table was generated for the STK scar area shapefile, where the areas of all disturbed features were summed for each RivID value.

12) Repeated step 9 for the STK debris area shapefile.

13) Joined the summary tables, from step 9 and 10, to the stream network based on the RivID field.

14) Using RivEX, assigned Strahler Order to stream segments and performed upstream accumulations for both the scar and debris areas. For the abandoned channel, accumulation values were manually re-set to 0.

15) Using the 'Feature to Point' tool in ArcMap, generated points for each stream segment affected by STK.

16) Scar and debris areas were summarized by Strahler Order (excluding hydrological/stream segments within lakes.)

**Supplementary Method 2. Workflow for broad-scale downstream accumulation of effects.**

To quantify the number of hydrologic features (streams, lakes, and coastlines) affected by slope thermokarst features (STKs) and to propagate the downstream effects of STK (main text Sect. 2.5), within Arctic drainage from continuous permafrost, we used an inventory of hydrologic features affected by STK (Kokoszka and Kokelj, 2020), hydrologic data (streams, lakes and coastlines) from the 1:50,000 National Hydro Network (NHN) dataset (Natural Resources Canada, 2016), and RivEX 10.25 Software (Hornby, 2017). Data were compiled in ArcMap 10.6.

Due to the significant amount of hydrologic data required to propagate STK affects throughout the entire study basin, geoprocessing was completed on the basis of Water Survey of Canada sub-sub-drainage areas (NHN Work Unit), for a total of 68 Work Units. A complete list of NHN Work Units is available from Kokoszka and Kokelj, 2020. Because the fluvial network was connected across NHN Work Units, accumulation analyses were first conducted for Work Units located in the headwaters of the study basin. Accumulation analyses were then conducted within downstream NHN Work Units to ensure propagation of STK effects throughout the entire study basin. The workflow involved the following steps:

1) For a specified Work Unit, NHN Primary Directed Network Linear Flow (PDNLF), Waterbody, and Littoral feature classes were imported to ArcMap to represent streams, lakes, and coastlines, respectively.

2) Re-projected the PDNLF, Waterbody, and Littoral feature classes to the Canada Lambert Conformal Conic projected coordinate system (CLCC).

3) Removed features from the Waterbody feature class that were attributed as watercourses or intermittent.

4) Modifications to hydrologic data were made for the specified NHN Work Units:

   a) NHN Work Unit 10TB002: Removed a littoral segment that extended beyond the coastline into the Arctic Ocean (nid = 4bcd3316c65c449a99f307803d0bddb3).

   b) NHN Work Unit 10MD002: Clipped hydrologic data to the reduced extent of the Work Unit delineation (Kokoszka and Kokelj, 2020).

   c) NHN Work unit 10MC002: Re-routed the Mackenzie River to propagate STK effects through the eastern portion of the Mackenzie Delta as opposed to the central region of the Mackenzie Delta by adding a PDNLF polyline (nid = 10mc002ADD01) and removing a PDNLF polyline (nid = 874e0d518ce34245a367869ddcbadeab).

5) Joined the attribute tables from the PDNLF feature class and STKI_Stream feature class (Kokoszka and Kokelj, 2020) based on the 'nid' fields.

6) Added an attribute field (ValDirect) to the PDNLF feature class with the field type assigned as 'short integer'.

7) Selected PDNLF features where the 'STK' field was equal to 1 (directly affected by STK) and from the selected PDNLF features, attributed the 'ValDirect' field with a value of 1.

8) Constructed a topological network and ran 'Quality Control' tools using RivEX. Error logs were generated by RivEX and were inspected for quality control. For this project, the error logs generated by RivEX included:

a) Small polylines composed of two vertices: Small polylines (< 1m) in length. Small polylines do not affect the integrity of the topological network. As such, for the purpose of this project, small polylines were not adjusted during the quality control process.

b) Polyline spikes: A digitization error where a single vertex is out of place that creates an acute angle along the length of the polyline. Spikes do not affect the integrity of the topological network, but can increase the length of a polyline segment. For the purpose of this project, polyline spikes were not adjusted because the increase in polyline length was deemed negligible compared to the overall length of polylines within the study basin.

9) Computed the upstream accumulation of direct STK affects, within the NHN Work Unit. In RivEX, selected 'attribute network' -> 'accumulate attribute in network' -> 'Run Tool' -> selected the variable to accumulate as 'ValDirect' -> specified the output field as 'AccDirect' -> selected 'OK'.

10) Identified indirectly affected PDNLF polylines (segments) (i.e. segments located upstream of directly affected segments). In ArcMap, started an edit session -> from the 'selection' tab, selected 'select by attribute' -> set selection method as 'create new selection' -> from the PDNLF layer, selected features where the 'AccDirect' field was > 0 -> selected 'OK' -> attributed the selected features 'STK' field with a value of 2 (indirect).

11) Identified PDNLF segments that were directly and indirectly affected by STK.  In ArcMap, started an edit session -> from the 'selection' tab, selected 'select by attribute' -> set the selection method as 'create new selection' -> from the PDNLF layer, selected features where the 'AccDirect' field was > 0 -> selected 'OK' -> started a new selection from the 'selection' tab by selecting 'select by attribute' -> set selection method as 'from the current selection' -> from the PDNLF layer, selected features where the 'STK' field  was not 0 -> selected 'OK' -> attributed the selected features 'STK' field with a value of 3 (both).

12) In ArcMap, visually inspected the attribution of the "STK" field, for PDNLF segments, by adjusting the colour scheme of the 'STK' field as follows:  'STK' = 0 to Cretan Blue (no STK affect), 'STK' = 1 to Mars Red (direct STK affect), 'STK' = 2 to Electron Gold (indirect STK affect), and 'STK' = 3 to Tuscan Red (both).

13) Identified indirectly affected Waterbody features (i.e. located upstream of directly affected PDNLF segments). In ArcMap, started an edit session -> from the 'selection' tab, selected 'select by attribute' -> set the selection method as 'create new selection' -> from the PDNLF layer, selected features where the 'AccDirect' field was > 0 and the 'Pseudo' field was equal to  0 -> selected 'OK' -> started a new selection from the 'selection' tab by selecting 'select by location' -> set the selection method as 'select features from' -> set the target layer as the Waterbody layer -> set the source layer as the PDNLF layer and ensured the 'use selected features' radio button was checked -> set the selection method as 'contain the source layer' -> selected 'OK' -> attributed the selected features 'STK' field with a value of 2 (indirect).

14) Identified Waterbody features that were directly and indirectly affected by STK. In ArcMap, started an edit session -> from the 'selection' tab, selected  'select by attribute' -> set selection method as 'create new selection' -> from the

PDNLF layer, selected features where the 'AccDirect' field was > 0 and the 'Pseudo' field was equal to 1 -> started a new selection from the 'selection' tab by selecting 'select by location' -> set the selection method as 'select features
from' -> set the target layer as the Waterbody layer -> set the source layer as the PDNLF layer and ensured the 'use selected features' radio button was checked -> set the selection method as 'contain the source layer' -> selected 'OK' -> attributed the selected features 'STK' field with a value of 3 (both).

15) In ArcMap, visually inspected the attribution of the "STK" field of Waterbody features by adjusting the colour scheme of the 'STK' field as follows:  'STK' = 0 to Cretan Blue (no STK affect), 'STK' = 1 to Mars Red (direct STK
affect), 'STK' = 2 to Electron Gold (indirect STK affect), and 'STK' = 3 to Tuscan Red (both).

16) In ArcCatalog, generated a new point feature class (AccSink) and added an attribute field (ValDirect) with the field type assigned as 'short integer'.

17) Locations where stream and lake derived STK affects propagated to the coastline were identified by visually inspecting the downstream path of PDNLF segments based on the 'STK' field's colour scheme (i.e. PDNLF
segments that were not Cretan Blue) and mapping AccSink points at the downstream end of PDNLF segments, that terminated at the coastline. In order to represent the number of upstream (accumulated) hydrologic features directly affected by STK at the coastline, the value of the 'AccDirect' field, from the PDNLF segment, was attributed to the 'AccDirect' field of the AccSink point.

18) Because PDNLF segments represent the main route of network flow, in some cases streams affected by STK did not
propagate to the main stem of the Mackenzie River (NHN Work Unit: 10LC000). In such cases, the corresponding value of the 'AccDirect' field was added to the AccSink point at the outlet of the Mackenzie River main stem. Similarly, values of the 'AccDirect' fields, for streams located near the outer reach of the Mackenzie Delta, were consolidated to the AccSink point located at the outlet of the Mackenzie River main stem.

19) Generated a 500m buffer (AccSinkBuffer) around the AccSink points.

20) Identified indirectly affected Littoral features (i.e. features located downstream of directly and indirectly affected PDNLF segments). In ArcMap, started an edit session -> from the 'selection' tab, selected 'select by location' -> set the selection method as 'select features from' -> set the target layer as the Littoral layer -> set the source layer as the AccSinkBuffer layer -> set the selection method as 'intersect the source layer' -> selected 'OK' -> started a new selection from the 'selection' tab by selecting 'select by attribute' -> set the selection method as 'select from current
selection' -> from the Littoral layer, selected features where the 'STK' field was equal to 0 -> selected 'OK' -> attributed the selected features 'STK' field with a value of 2 (indirect).

21) Identified Littoral features that were directly and indirectly affected by STK. In ArcMap, started an edit session -> from the 'selection' tab, selected 'select by location' -> set the selection method as 'select features from' -> set the target layer as the Littoral layer -> set the source layer as the AccSinkBuffer layer -> set the selection method as
'intersect the source layer' -> selected 'OK' -> started a new selection from the 'selection' tab by selecting 'select by attribute' -> set selection method as 'select from current selection' -> from the Littoral layer, selected features where the 'STK' field was equal to 1 -> selected 'OK' -> attributed the selected features 'STK' field with a value of 3 (both).

22) Repeated steps 1 to 21 for each NHN Work Unit. Where STK affects propagated between NHN Work Units, prior to conducting the accumulation analysis the value of the 'AccDirect' field, for the upstream PDNLF segment (i.e. located in the upstream NHN Work Unit), was attributed to the 'AccDirect' field for the downstream PDNLF segment (i.e. located in the downstream NHN Work Unit).

23) For each NHN Work Unit, segments from the PDNLF feature class, where the 'STK' field was equal to 1 (i.e. directly affected by STK), were selected and exported as new feature classes. The feature classes were then merged to generate a downstream trace that represented the propagated and accumulated STK affects across the entire study basin.

24) The downstream trace was visually inspected to ensure continuity of STK affects between NHN Work Units.

**Supplementary Method 3. RivEX Workflow to Assess Strahler Order**

To derive the Strahler stream and lake orders within four watersheds (Keele/Redstone, Peel, Amundsen Gulf, and Banks Island) of the Arctic drainage area from continuous permafrost of northwestern Canada (main text Sect. 2.5), we used hydrologic data (streams and lakes) from the 1:50,000 National Hydro Network dataset (NHN) (Natural Resources Canada,

2016) and RivEX 10.25 Software (Hornby, 2017). Data were compiled in ArcMap 10.6. Steps in the workflow included:

1)   In ArcMap, imported NHN Primary Directed Network Linear Flow (PDNLF), Waterbody, and NHN WorkUnit feature classes to represented streams, lakes, and hydrologic boundaries, respectively.

2)   Generated watershed delineations by merging the following NHN Work Unit feature classes for each specified watershed:

a)   Keele/Redstone: 10HA000 and 10HB000

b)   Peel: 10MA000, 10MB000, and 10MC002

c)   Amundsen Gulf: 10OA001, 10OA002, 10OB000, 10OC001, and 10OC002

d)   Banks Island: 10TA001, 10TA002, 10TA003, 10TA004, 10TB001, and 10TB002

3)   Modifications to hydrologic data were made for the specified NHN Work Units:

a)   NHN Work Unit 10TB002: Removed a littoral feature that extended beyond the coastline into the Arctic Ocean (nid = 4bcd3316c65c449a99f307803d0bddb3).

b)   NHN Work Unit 10MC002: Work Unit feature class was manually reduced to specify the drainage area from a specified segment of the Peel River (nid = f1d3cf54069c4e4f9f35ddef517e4b9d). The 10MC002 Work Unit area included PDNLF polylines within the Mackenzie Delta. Computing Strahler order for

PDNLF polylines within the 10MC002 Work Unit would have required computing Strahler order for the entire Mackenzie River, which was not feasible given computational restraints.

4)   Merged PDNLF and Waterbody feature classes for their respective watersheds (see step 2). The merged PDNLF and Waterbody feature classes, from the Peel watershed, were clipped to the reduced watershed delineation (see Step 3b).

5)   Constructed a topological network for each watershed using the merged PDNLF feature classes (see Step 4), and ran 'Quality Control' tools, using RivEX. The quality control error logs included small polylines and spikes within the networks. However, the presence of small polylines and spikes do not affect the topology of the network and were disregarded in the quality control process.

6)   Computed Strahler Order for PDNLF polylines by selecting 'Strahler' from the 'Network Attribution" tool in RivEX.

7)   Performed a spatial join between PDNLF polylines (join features) and Waterbody polygons (target features) to determine lake order. The spatial join matched PDNLF polylines that were located within Waterbody features based on the maximum Strahler Order of the PDNLF polylines, using a one-to-one join operation.

8)   Visually inspected the Strahler stream and lake order outputs for quality assurance.

**Supplementary Figures 1-4**

[Figure]

[Figure]

**Figure S1.** Retrogressive thaw slumps and debris tongues showing transects for the topographic profiles in Figure 2g. The
terrain models are from UAV surveys described in Sect. 2.2, 2.3 and summarized in Table S2.

**Figure S2.** Frequency distributions for thaw slump (a) area, (b) volume and (c) maximum concavity depth grouped by downstream connectivity based on slump digitization of 2011 LiDAR. Class 0 indicates no physical connection between the slump and downstream environment; 1 is a physical connection between an active or bare scar and the downstream environment; and 2 is evidence of downstream deposition, which is expressed as a debris tongue in a valley bottom, or a sediment lobe protruding into an adjacent lake or coastline.

[Figure]

**Figure S3.** Willow River main stem view looking southwest. Photograph shows small shallow slides in foreground, and large deep-seated translational failures which have evolved into retrogressive thaw slumps. Slide materials have runout onto the braided floodplain. Numerous retrogressive thaw slumps are visible in background on slopes of incised tributary streams. Photograph is upstream view towards inset (b) on Figure 6.

**Figure S4.** Panels show downstream effects of slope thaw-driven mass wasting and some oblique photographs of slope disturbances. Examples of thaw-driven increases in downstream sedimentation include examples from: i. Sachs River inflow to Big Fish Lake, Banks Island; ii. Inflow of Miner and Kugalik Rivers to Husky Lake estuary, Mackenzie Delta region; iii. Development of massive, deep seated translational failure on Johnson River, central Mackenzie Valley. Supporting oblique photographs of thaw-driven sediment sources are provided for ii and iii. Locations of i, ii and iii are indicated on Figure 8.

[Figure]

**i.** Sachs River and Big Fish Lake, Banks Island (a) 1985 and (b) 2018 (Lat/long: 71.8237 N, -124.4983 W). (c)  August 1, 2015 photograph of a large retrogressive thaw slump located on a Sachs River tributary about 35 km upstream of Big Fish Lake. (d)  inflow of Sachs River into Big Fish Lake during summer baseflow conditions (August 1, 2015), showing slump derived turbidity.

[Figure]

**ii.** Inflow of Miner River (lower left) and Kugalik River (lower right) into Husky Lake estuary (a) 1991 and (b) 2018. Note turbidity of Miner River and estuary in 2018 (Lat/long: 69.1617 N, -131.0135 W). (c) Large retrogressive thaw slumps on upper sections of Miner River and effects of thaw-driven slope to stream sediment delivery. (d) shows dark, DOC-rich water of undisturbed Miner River on left, and turbid conditions downstream (to the right) of the debris tongues which have blocked the channel. Photographs are from summer 2020.

[Figure]

**iii.** Johnson River near the confluence with Mackenzie River showing changes caused by a major deep-seated translational failure that occurred in fall 2017. Images show the area in (a) 1991 and (b) following major slide development (2018) (Lat/long: 63.6587 N, -124.0530 W). Arrows on (b) are 2 km in length and show the upper part of the slide and the debris deposit. Note the high turbidity  the river downstream of the landslide and the development of an alluvial deposit at Johnson River confluence with the Mackenzie River in the upper right. (c, d)  southward oblique views showing the enormous debris flow deposit that has infilled the Johnson River valley. The fine-grained, ice-rich permafrost materials that have been translocated into the valley are melting out causing thermokarst and continued mobilization of materials. For scale, note forest cover on the blocks of permafrost debris that slid along a 3-5° slope into the valley. Blockage of Johnson River caused an upstream lake to form, and the river has incised a canyon through the debris deposit.

**Supplementary Tables 1-5**

**Table S1.** Summary of research  the spatial datasets and related methods sections, geographic locations, and sources of the datasets used in this multi-scale study.

| Research  | Base datasets | Geographic locations (Figure 1) | Data source |
|---|---|---|---|
| (A) Thaw-driven geomorphic change and slope to stream connectivity | *Sect. 2.2 & 2.3*; 2011 LiDAR ( 1 m DEM) | Peel Plateau; Anderson Plain & Tuk Coastlands | NWT Centre for Geomatics, Gov. of Northwest Territories |
| | *Sect. 2.2 & 2.3*; UAV imagery (≤ 3 cm, downsampled to 1 m DEM) (Table S2) | Peel Plateau | van der Sluijs et al., 2018; NWT Centre for Geomatics, Gov. of Northwest Territories |
| (B) Medium scale catchment effects of thaw-driven mass wasting | *Sect. 2.4*; 2016-2017 Sentinel2 orthomosaic, 10m | Peel Plateau; SE Banks Island | Rudy et al., 2020 |
| | *Sect. 2.4*; 1986, 2002, 2018 Landsat, 30 m | Willow River | Rudy and Kokelj, 2020 |
| | *Sect. 2.4*; Stream sediment and discharge data, Peel watershed | Peel Plateau | Shakil et al., 2020b |
| (C) Distribution of mass-wasting effects on fluvial networks and propagation across Arctic drainage from continuous permafrost of Northwestern Canada | *Sect. 2.5*; SPOT 4/5 (2004-2010), 15m; Sentinel-2 (2016, 2017), 10m | Arctic drainage from continuous permafrost & Banks, Amundsen, Peel, Keele-Redstone watersheds | Kokoszka and Kokelj, 2020 |
| | *Sect. 2.5*; National Hydro Network dataset | | NRCan, 2016 |

**Table S2.** Summary of UAV surveys conducted over the four-year campaign to support Sect. 3.1-3.3 (modified from Van der Sluijs et al., 2018).

| Site | Date | UAV | Res (cm) | Area (ha) | Photos (no.) |
|---|---|---|---|---|---|
| **D1** (67.1771° N -135.7555° W) | 2015-07-28 | PX8 | 1.5 | 4.3 | 295 |
| | 2015-07-29 | PX8 | 1.5 | 3.8 | 291 |
| | 2016-08-03 | Inspire | 1.3 | 1.7 | 79 |
| | 2017-07-27 | P4P | 1.2 | 6.3 | 316 |
| | 2018-09-18 | eBee | 1.5 | 9.0 | 311 |
| **FM3** (67.2539° N -135.2732° W) | 2015-07-29 | PX8 | 1.9 | 28.3 | 658 |
| | 2016-08-02 | Inspire | 2.4 | 34.3 | 583 |
| | 2017-07-26/28 | eBee | 3.3 | 365.0 | 3,499 |
| | 2018-09-19 | eBee | 3.3 | 88 | 463 |
| **FM2** (67.2545° N -135.2286° W) | 2016-08-02 | Inspire | 2.4 | 83.6 | 1516 |
| | 2017-07-26/28 | eBee | 3.3 | 365.0 | 3499 |
| | 2018-09-18 | eBee | 3.3 | 161 | 982 |
| **Husky** (67.5207 ° N -135.3005 W) | 2016-08-03 | Inspire | 1.7 | 36.8 | 773 |
| | 2017-07-28 | eBee | 3.4 | 100.5 | 723 |
| | 2018-09-19 | eBee | 3.2 | 90.0 | 510 |
| **CB** (67.1814° N -135.7295° W) | 2015-07-28 | PX8 | 1.5 | 12.1 | 711 |
| | 2018-09-18 | eBee | 2.7 | 28.1 | 438 |
| Mean | | | 2.3 | 83 | 920 |
| Stdev | | | 0.8 | 115 | 1024 |
| Sum | | | | 1,053 | 15,647 |

Note: Flights conducted for thermal mapping, oblique still photography and video purposes are not included. Table headings "Res" is resolution or pixel size. UAV platforms: Spyder PX8 Plus (PX8), Phantom 2 Vision Plus (P2), RX4-S Surveyor (RX4), Inspire 1 Pro (Inspire), eBee Plus RTK/PPK (eBee), and Phantom 4 Pro (P4P).

**Table S3** Summary statistics of slump area, volume and scar concavity depth estimates for the Peel Plateau, and Anderson Plain and Tuktoyaktuk Coastlands 2011 LiDAR Corridor (n=71).

| | | Area (m$^2$) | Volume (m$^3$) | Concavity depth (m) | |
| --- | --- | --- | --- | --- | --- |
| | | | | Mean | Maximum |
| Mean | | 14,934 | -106,003 | -1.82 | -4.76 |
| Median | | 3,323 | -3,860 | -1.18 | -3.36 |
| Std. Deviation | | 38,056 | 480,480 | 2.27 | 4.73 |
| Skewness | | 5 | -6 | -3.62 | -3.34 |
| Kurtosis | | 26 | 45 | 15.46 | 12.85 |
| Minimum | | 242 | -3,653,390 | -14.39 | -28.37 |
| Maximum | | 253,937 | -128 | -0.11 | -1.48 |
| Sum | | 1,060,280 | -7,526,216 | | |
| Percentiles | 25 | 1,495 | -13,982 | -2.10 | -5.47 |
| | 50 | 3,323 | -3,860 | -1.18 | -3.36 |
| | 75 | 7,730 | -995 | -0.70 | -2.22 |

**Table S4.** Summary statistics for size metrics of active thaw-slumps in the Willow River catchment for 1986, 2002 and 2018. Area estimates were determined by digitizing orthorectified, color balanced,  free Landsat imagery (Rudy and Kokelj, ). Thaw slump volume and maximum concavity depth were estimated using relationships shown in Figure 4.

| Parameter | 1986 | 2002 | 2018 |
|---|---|---|---|
| **Scar area (m$^2$)** | | | |
| Count | 21 | 73 | 198 |
| Cumulative area | 105,081 | 669,249 | 3,545,445 |
| Median | 4,512 | 5,688 | 8,293 |
| Mean (STdev) | 5,004 (2,601) | 9,768 (10,608) | 17,906 (29,685) |
| Max | 13,497 | 63,838 | 198,986 |
| Min | 1,649 | 766 | 1,073 |
| | | | |
| **Scar volume (m$^3$)** | | | |
| Cumulative volume | 144,868 | 1,437,340 | 11,688,197 |
| Median | 5,537 | 7,687 | 13,119 |
| Mean (STdev) | 6,898 (5,513) | 19,690 (35,515) | 59,031 (161,741) |
| Max | 26,154 | 236,479 | 1,184,216 |
| Min | 1,330 | 449 | 723 |
| | | | |
| **Concavity depth (m)** | | | |
| Median | 3.6 | 3.9 | 4.5 |
| Mean (STdev) | 3.6 (0.67) | 4.1 (1.5) | 5.1 (2.1) |
| Max | 5.3 | 9.3 | 14.1 |
| Min | 2.5 | 1.9 | 2.1 |
| | | | |
| **Debris tongue area (m$^2$)** | | | |
| Count | 6 | 31 | 67 |
| Median | 5,046 | 6,165 | 8,884 |
| Mean (STdev) | 18,655 (24,511) | 9,681 (10,071) | 15,394 (18,857) |
| Max | 59,011 | 39,682 | 100,569 |
| Min | 1,631 | 1,250 | 1,087 |

\* Landsat dates: 1986_07_07; 2002_07_20; 2018_07_22

**Table S5.** Results Dunn's post-hoc test for comparison of thaw slump scar area between 1986, 2002 and 2018 following a Kruskal-Wallis Rank Sum test (p>0.00001).

| Comparison dates | Z score | P. unadjusted | P. adjusted |
|---|---|---|---|
| 2002-2018 | -3.4614 | 0.0005 | 0.0008 |
| 1986-2002 | 1.4714 | 0.1411 | 0.1411 |
| 1986-2018 | 3.6529 | 0.0002 | 0.0008 |

***Dunn's post-hoc results indicate significant differences between 1986 and 2018, and 2002 and 2018.

**Supplementary Videos 1-3**

**Video S1.**
Video shows truncation of a small tundra lake by thaw-slump growth causing rapid drainage on the Peel Plateau in 2015 (67.5207 ° N -135.3005° W).
https://www.nwtgeoscience.ca/services/permafrost-thaw-slumps/video-permafrost-thaw-causes-lake-drainage-peel-plateau-nwt

**Video S2.**
UAV fly through of a large active retrogressive thaw slump in Willow River catchment shown in Figure 6b (68.1119° N -135.6806° W).
https://www.nwtgeoscience.ca/services/permafrost-thaw-slumps/drone-survey-permafrost-mega-slump-willow-river-nwt

**Video S3.**
Animated GIF image depicting a Landsat time-series animation covering the 1985-2019 period for lower Willow River catchment on the eastern edge of the Mackenzie Delta showing evolution of major thaw slumps most notably in lower left corner of imagery (see Fig. 6b). The time series shows acceleration of disturbance activity in the late 1990s and early 2000s, 440 alluviation of the Willow River Channel and infilling of the large Mackenzie Delta lake from 2007 and 2019 with slump-derived sediment transported by the Willow River (Fig. 6c). Animation was generated using two ©Google Earth Engine applications: 1) the "LandTrendr time series animation app, URL: https://emaprlab.users.earthengine.app/view/lt-gee-time-series-animator" to generate the animation, and 2) the "Snazzy EE-TS-GIF, URL: https://jstnbraaten.shinyapps.io/snazzy-ee-ts-gif/" for annotations. The Landsat time series has been smoothed by LandTrendr spectral-temporal segmentation on 445 ©Google Earth Engine (Kennedy et al., 2018). URL: https://emapr.github.io/LT-GEE/ui-applications.html  (Accessed 3 August 2020).

---

## Author Response (AR2)

April 6, 2021

Dear Peter,

Thank you very much for your helpful comments on the paper and we greatly appreciate you recommending this as a highlight paper. The following document indicates how we have implemented the minor revisions that you have suggested.  Our track changed documents and PDF versions are attached.

Please let us know if you require further information. We look forward to hearing from you in due course.

Best wishes,

Steve

**Author Reply to Editor's Minor Revisions.**

Minor track change edits were implemented throughout the text.

P2 L62: Regarding use of renewal: I think renewal does work, however we have followed your suggestion and adjusted this to "rejuvenation" which is also an adjective we have used in the past. There is evidence that throughout the Holocene, and most prominently in the early Holocene, streams have transported lots of post-glacial sediments from slopes into downstream systems. However, thaw-driven processes have certainly been "invigorated" by climate-driven thaw. Adjustment also made on P31 L807.

P7 L176: We moved reference to the nature of biophysical conditions in southern environments to the front of the clause following the sequence presented for low and high Arctic descriptions and we also removed some redundant words. As I understand this follows your suggestion. Regardless, the flow has been improved.

P10 L291: Suggested editorial revision implemented.

Table 1: Suggested editorial revisions implemented.

Figure 2: Suggested editorial revision implemented.

P19 L515: Suggested editorial revision implemented.

P20 L550, 552: Suggested editorial revisions implemented.

Figure 6: Please note that in this figure we are portraying accumulations as a function of disturbance area, not as disturbance count. The impacts to the stream reaches with no apparent dot reveal that disturbance area is very small (less than 1 ha, so very small slide(s) or slump(s)). In this way, the reader

can determine that the reach is impacted but total disturbance area for that stream reach is less than 1 ha. This is explained in the caption.

P23 L582: We have used Figs 6, S3 as examples because Willow River drains into Peel Channel, and in our broad scale accumulation, Peel watershed effects are also routed through Peel Channel. However, we have adjusted text slightly for clarification.

Some of the main Peel River tributaries are increasingly being affected by large slope thermokarst failures and shallow slides "similar to those observed in the Willow River catchment" (Figs. 6, S3).

P23 L593: Source of the compilation is added here (Shakil et al., 2020b).

Note that we have reviewed the manuscript to ensure figure references appropriately match figures. The edits you have provided in track changes throughout the manuscript and supplement have been implemented.

Supplementary materials:

P2 L45-50: "nid" was defined as suggested.

P3 L85-86: Point 10 was simplified to read "In order to ensure the consistency of areal estimates, scar and debris tongue areas were computed using the same projected coordinate system used in Rudy and Kokelj (2020)."

P4 L115: Step 2 was removed and included in the Method 2 introductory paragraph, which is consistent with how this procedural step was addressed in Method 1.

P5 Step 12 and 15, with reference to adjusting color for visual inspection. These sections were integrated and simplified, and reference to particular color scheme was removed.

Figure S1. Caption adjusted.

Figure S3. Photograph date added.

Figure S4iii. Photograph adjusted to provide a different view of the Johnson River slide.

Tables: Minor adjustments implemented.

Videos: hyperlinks and access dates added.

---

## Author Response (AR3)

April 19, 2021

Dear Peter,

Thank you for carefully reviewing our manuscript.  The following document indicates how we have implemented the minor revisions that you have suggested.  Our track changed documents, and PDF versions are attached. We feel that the Manuscript has been improved and that is now grammatically consistent.

Please let us know if you require further information. We look forward to hearing from you in due course.

Best wishes,

Steve

**Author Reply to Editor's Minor Revisions.**

Minor track change edits were implemented throughout the text. The suggested edits were not implemented in only one instance where the change would have modified the intended meaning of the sentence. Serial commas were added throughout the manuscript and to the supplementary material text.

The narrative has been clarified to indicate that the study area includes both continuous and discontinuous permafrost terrain. The source of the NHN data and Watershed areas are briefly described and the Supplementary Methods are specifically referenced to direct readers to the sources of the NHN data and to explanation of how study "watersheds" and study "area" were constructed.

P4 L 119-120. We have adjusted the text to address the Editor's comment that the study area includes continuous and discontinuous permafrost areas.

Revised text:

"Here we present a suite of spatially nested case-studies bounded by Arctic drainage from permafrost terrain of northwestern Canada (Fig. 1). The study area is predominantly in the zone of continuous permafrost however, we also include the Great Bear River drainage and a few other Mackenzie River tributaries that drain northern margins of the extensive discontinuous permafrost zone."

P5 L145-146. In this study, we defined the broadest area of investigation as "Arctic drainage from permafrost terrain of northwestern Canada" which is the $10^6$ km$^2$ scale. The second scale of inquiry is at the $10^5$ km$^2$ area, which we refer to as "watersheds" so that a reader can distinguish analyses and

results pertaining to different scales of inquiry. Regarding the Editor's comment, we have modified the text slightly at line 148, to clarify that these "watersheds" are composite drainage areas. We also provide brief elaboration in the methods and would note that definitions and the NHN work units (sub sub drainage areas) are explicitly defined for the 3 scales of inquiry in supplementary methods 1-3. Supplementary methods are also referenced now early in the Manuscript to direct a reader there for more information.

Revised text:

"To assess the potential propagation of watershed-scale slope thermokarst effects ($10^4$ to $10^5$ km2) (goal C), disturbance distribution was analyzed within a Strahler-order framework for composite drainage areas characterized by contrasting terrain and permafrost conditions that we defined for this study as the Banks Island, Amundson Gulf, Peel River and Keele-Redstone watersheds (Fig. 1, Table S1)."

Figure 1. We have changed the label Keele/Redstone Watershed to "Keele-Redstone Watersheds" and the legend title "Discontinuous permafrost extent" to "Southern limit of continuous permafrost". The source of the catchment, watershed and drainage boundaries is now indicated in the caption.

In multiple locations, including the caption, we now clarify that the Watershed scale of inquiry was defined by compositing sub-sub drainage areas in the NHN dataset, including earlier reference to the Supplementary Methods.

Figure.1. Study region map showing the distribution and dominant geomorphic environments affected by thaw-driven mass wasting, and the locations and scales of investigation constrained by the 994,860 km$^2$ area of Arctic drainage from permafrost terrain of northwestern Canada. Fine-scale thaw slump mapping utilizing high-resolution UAV and LiDAR terrain models is indicated by the orange corridors (Peel Plateau; Tuktoyaktuk Coastlands and Anderson Plain); small to medium scale catchments including Willow River, Peel Plateau and southeastern Banks Island areas are indicated by red polygons; focal watersheds are outlined in blue, and the Arctic drainage study area is shaded in grey. The disturbance data on the map are adapted from Segal et al., 2016b and Kokelj et al., 2017a. Late-glacial limit is from Dyke and Prest, 1987, bedrock geology is from Fulton, 1995, and the permafrost boundary is from Brown et al., 1997. The Willow River Catchment is part of a sub-sub-drainage, and the Watershed and Study area boundaries are composites of several sub-sub-drainage areas (Methods S1-3) from the National Hydro Network (NHN) geodatabase (Natural Resources Canada, 2016). The base map is from ESRI ArcGIS.

P7L172. We have adjusted the narrative to clarify that our study area also comprises some drainages from the northern zone of extensive discontinuous permafrost. We hope this adjustment is sufficient to address the editor's concerns.

"This broadest scale of inquiry is defined as the Arctic drainage networks of northwestern Canada, primarily from continuous permafrost and the northern limits of extensive discontinuous permafrost

(Fig. 1). The 994,860 km$^2$ Arctic drainage area of northwestern Canada is comprised of 68 sub sub drainage areas defined in the National Hydro Network geodatabase (Natural Resources Canada, 2016). This study area is characterized by a diversity of permafrost, geological, climate, and ecosystem conditions (Fig. 1)."

P10L280-282. To address Editor's comments, we have made changes to section 2.1 and to the caption in Figure 1 to define our study area. We have also adjusted the text in this location to read:

"Here we summarize methods to identify individual segments of the hydrological network affected by active thaw-driven mass wasting and the framework to map the propagation pathways of effects through watersheds for Arctic drainage from permafrost terrain of northwestern Canada (Fig. 1)."

P11L321-322. Text was modified to again clarify that the "watershed study areas" are composed of several sub-sub-drainage areas.

"To summarize information on the distribution of watershed effects, Strahler Order was computed (Methods S3) for the 4 major watershed-scale study areas of Banks Island (70,794 km2), Amundsen Gulf (90,288 km2), Peel River (76,506 km2), and Keele-Redstone (39,957 km2) (Fig. 1). For each major watershed, the respective Work Units (sub-sub-drainage areas) and PDNLF Shapefiles were merged and the 'Strahler Order Tool' in RivEx 10.25 was used to compute Strahler Order for each PDNLF polyline. For each major watershed study area, directly and indirectly affected streams and lakes were summarized by Strahler Order."

Figure 8. Editorial suggestions to adjust labels to "Keele-Redstone Watersheds" and "Southern limit of continuous permafrost extent" were implemented.

The Figure 8 caption was adjusted to address the Editors comments to indicate what the NHN abbreviation stands for, and to cite the source of the NHN Database.

"Figure 8. Thaw-driven landslide density and downstream accumulation of effects through Arctic drainage from permafrost terrain of northwestern Canada. Heat map depicts all directly affected stream, lake, and coastal National Hydro Network (NHN) segments mapped. All upstream accumulation values shown within the fluvial network are >2, and at the coast are >9. Accumulated effect contributions of the Mackenzie and Peel Rivers to the Beaufort Sea are routed separately for comparison. (a) Counts of direct and accumulated thaw-driven mass wasting effects to fluvial systems partitioned by Strahler order and hydrological feature type for the four major watersheds outlined in blue. (b) Table showing lengths of directly affected hydrological network and accumulated effects, count of directly and indirectly affected lakes and total directly affected coastline for western Arctic drainage from permafrost terrain. Remote sensing examples of thaw-driven downstream sedimentation provided in Fig. S4 for: i) Sachs River and Fish Lake, ii) Miner River inflow to the Husky Lakes estuary and iii) massive deep-seated permafrost failure on Johnson River. Late glacial limit is from Dyke and Prest, 1987, bedrock geology is from Fulton, 1995, the permafrost boundary is from Brown et al., 1997, and the National Hydro Network (NHN) base data is from Natural Resources Canada, 2016. Base map is from ESRI ArcGIS Online."

---

## Author Response (AR4)

May 28, 2021

Dear Peter,

Thank you for again carefully reviewing the manuscript and identifying several technical corrections which we have implemented. We also note that numbers of impacted stream lengths, lakes and coastline in Figure 8 as well as references to those numbers in the manuscript text have been updated to reflect the finalized dataset that is being concurrently published in Kokoszka and Kokelj, 2021. The adjustments were necessary due to the accidental double counting of some impacted NHN segments in one sub-sub-catchment caught during a rerunning of the flow accumulation analysis, and also due to the need to exclude PDNLF segments through lakes from the total length of impacted stream length calculations. These updates have no effect on the main results, interpretations or conclusions made in this paper.

Please let us know if you require further information. We look forward to hearing from you in due course.

Best wishes,

Steve